# CLAIRVOYANCE:
# A PIPELINE TOOLKIT FOR MEDICAL TIME SERIES

**Daniel Jarrett**\*
University of Cambridge, UK
daniel.jarrett@maths.cam.ac.uk

**Jinsung Yoon**\*
Google Cloud AI, Sunnyvale, USA
University of California, Los Angeles, USA
jinsungyoon@google.com

**Ioana Bica**
University of Oxford, UK
The Alan Turing Institute, UK
ioana.bica@eng.ox.ac.uk

**Zhaozhi Qian**
University of Cambridge, UK
zhaozhi.qian@maths.cam.ac.uk

**Ari Ercole**
University of Cambridge, UK
Cambridge University Hospitals NHS Foundation Trust, UK
ae105@cam.ac.uk

**Mihaela van der Schaar**
University of Cambridge, UK
University of California, Los Angeles, USA
The Alan Turing Institute, UK
mv472@cam.ac.uk

## ABSTRACT

Time-series learning is the bread and butter of data-driven *clinical decision support*, and the recent explosion in ML research has demonstrated great potential in various healthcare settings. At the same time, medical time-series problems in the wild are challenging due to their highly *composite* nature: They entail design choices and interactions among components that preprocess data, impute missing values, select features, issue predictions, estimate uncertainty, and interpret models. Despite exponential growth in electronic patient data, there is a remarkable gap between the potential and realized utilization of ML for clinical research and decision support. In particular, orchestrating a real-world project lifecycle poses challenges in engineering (i.e. hard to build), evaluation (i.e. hard to assess), and efficiency (i.e. hard to optimize). Designed to address these issues simultaneously, Clairvoyance proposes a unified, end-to-end, autoML-friendly pipeline that serves as a (i) software toolkit, (ii) empirical standard, and (iii) interface for optimization. Our ultimate goal lies in facilitating transparent and reproducible experimentation with complex inference workflows, providing integrated pathways for (1) personalized prediction, (2) treatment-effect estimation, and (3) information acquisition. Through illustrative examples on real-world data in outpatient, general wards, and intensive-care settings, we illustrate the applicability of the pipeline paradigm on core tasks in the healthcare journey. To the best of our knowledge, Clairvoyance is the first to demonstrate viability of a comprehensive and automatable pipeline for clinical time-series ML.

**Python Software Repository**: https://github.com/vanderschaarlab/clairvoyance

## 1 INTRODUCTION

Inference over time series is ubiquitous in medical problems [1–7]. With the increasing availability and accessibility of electronic patient records, machine learning for *clinical decision support* has made great strides in offering actionable predictive models for real-world questions [8, 9]. In particular, a plethora of methods-based research has focused on addressing specific problems along different stages of the clinical data science pipeline, including preprocessing patient data [10, 11], imputing missing measurements [12–16], issuing diagnoses and prognoses of diseases and biomarkers [17–25], estimating the effects of different treatments [26–31], optimizing measurements [32–36], capturing

---

\*Authors contributed equally

uncertainty [37–41], and interpreting learned models [42–46]. On the other hand, these component tasks are often formulated, solved, and implemented as mathematical problems (on their own), resulting in a stylized range of methods that may not acknowledge the complexities and interdependencies within the real-world clinical ML project lifecycle (as a composite). This leads to an often punishing *translational barrier* between state-of-the-art ML techniques and any actual patient benefit that could be realized from their intended application towards clinical research and decision support [47–51].

**Three Challenges**  To bridge this gap, we argue for a more comprehensive, systematic approach to development, validation, and clinical utilization. Specifically, due to the number of moving pieces, managing real-world clinical time-series inference workflows is challenging due the following concerns:

- First and foremost, the *engineering* problem is that building complex inference procedures involves significant investment: Over 95% of work in a typical mature project is consumed by software technicals, and <5% addressing real scientific questions [52]. As a clinician or healthcare practitioner, however, few resources are available for easily developing and validating *complete* workflows. What is desired is a simple, consistent development and validation workflow that encapsulates all major aspects of clinical time-series ML—from initial data preprocessing all the way to the end.

- Second, the *evaluation* problem is that the performance of any component depends on its *context*; for instance, the accuracy of a prediction model is intimately tied to the data imputation method that precedes it [13, 14]. As an ML researcher, however, current empirical practices typically examine the merits of each component individually, with surrounding steps configured as convenient for ensuring "all else equal" conditions for assessing performance. What is desired is a structured, realistic, and reproducible method of comparing techniques that honestly reflects interdependencies in the gestalt.

- Lastly, the *efficiency* problem is that sophisticated designs tend to be resource-intensive to optimize, and state-of-the-art deep learning approaches require many knobs to be tuned. As a clinical or ML practitioner alike, this *computational* difficulty may be compounded by pipeline combinations and the potential presence of temporal distribution shifts in time-series datasets [53]. What is desired is a platform on which the process of pipeline configuration and hyperparameter optimization can be automated—and through which new optimization algorithms to that effect may be built and tested.

**Contributions**  We tackle all three issues simultaneously. The Clairvoyance package is a unified, end-to-end, autoML-friendly pipeline for medical time series. (i) As a *software toolkit*, it enables development through a single unified interface: Modular and composable structures facilitate rapid experimentation and deployment by clinical practitioners, as well as simplifying collaboration and code-sharing. (ii) As an *empirical standard*, it serves as a complete experimental benchmarking environment: Standardized, end-to-end pipelines provide realistic and systematic context for evaluating novelties within individual component designs, ensuring that comparisons are fair, transparent, and reproducible. (iii) Finally, as an *interface for optimization* over the pipeline abstraction, Clairvoyance enables leveraging and developing algorithms for automatic pipeline configuration and stepwise selection, accounting for interdependencies among components, hyperparameters, and time steps. Through illustrative examples on real-world medical datasets, we highlight the applicability of the proposed paradigm within personalized prediction, personalized treatment planning, and personalized monitoring. To the best of our knowledge, Clairvoyance is the first coherent effort to demonstrate viability of a comprehensive, structured, and automatable pipeline for clinical time-series learning.

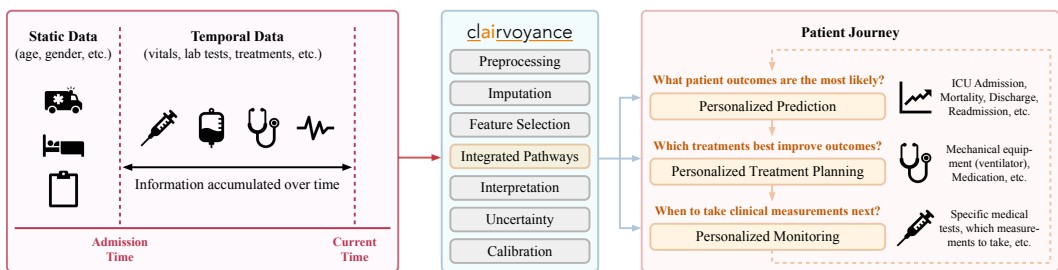

Figure 1: *Clairvoyance and the Patient Journey*. The healthcare lifecycle revolves around asking (1) *what* outcomes are most likely, (2) *which* treatments may best improve them, and (3) *when* taking additional measurements is most informative. Utilizing both static and temporal data, Clairvoyance provides corresponding pathways for personalized prediction of outcomes, personalized estimation of treatment-effects, and personalized monitoring.

## 2  THE CLAIRVOYANCE PIPELINE

**The Patient Journey**  Consider the typical patient's interactions with the healthcare system. Their healthcare lifecycle revolves tightly around (1) forecasting outcomes of interest (i.e. the prediction problem), (2) selecting appropriate interventions (i.e. the treatment effects problem), and (3) arranging followup monitoring (i.e. the active sensing problem). Each of these undertakings involve the full complexity of preparing, modeling, optimizing, and drawing conclusions from clinical time series. Clairvoyance provides model pathways for these core tasks in the patient journey (see Figure 1) —integrated into a single pipeline from start to finish (see Figure 2). Formally, these pathways include:

- *Predictions Path*. Let $\{(\mathbf{s}_n, \mathbf{x}_{1:T_n})\}_{n=1}^N$ denote any medical time-series dataset, where $\mathbf{s}_n$ is the vector of static features for the $n$-th patient, and $\mathbf{x}_{1:T_n} \doteq \{\mathbf{x}_{n,t}\}_{t=1}^{T_n}$ is the vector sequence of temporal features. *One-shot* problems seek to predict a vector of labels $\mathbf{y}_n$ from $(\mathbf{s}_n, \mathbf{x}_{n,1:T_n})$: e.g. prediction of mortality or discharge, where $y_n \in \{0, 1\}$. *Online* problems predict some target vector $\mathbf{y}_{n,t}$ from $(\mathbf{s}_n, \mathbf{x}_{n,1:t})$ at every time step: e.g. $\tau$-step-ahead prediction of biomarkers $\mathbf{y}_{n,t} \subseteq \mathbf{x}_{n,t+\tau}$.

- *Treatment Effects Path*. For individualized treatment-effect estimation [26–31], we additionally identify interventional actions $\mathbf{a}_{n,t} \subseteq \mathbf{x}_{n,t}$ at each time step (e.g. the choices and dosages of prescribed medication), as well as corresponding measurable outcomes $\mathbf{y}_{n,t} \subseteq \mathbf{x}_{n,t+\tau}$. The learning problem now consists in quantifying the (factual or counterfactual) potential outcomes $\mathbf{y}_{n,t+\tau}$ that would result from any specific sequence of interventions and patient covariates $(\mathbf{s}_n, \mathbf{x}_{n,1:t}, \mathbf{a}_{n,1:t})$.

- *Active Sensing Path*. In addition to mapping (already-measured) covariates to targets, the very decision of what (and when) to measure is also important under resource constraints. In medical settings, active sensing deals with balancing this trade-off between information gain and acquisition costs [32–36]. With reference to some downstream task (e.g. predicting $\mathbf{y}_{n,t+1}$), the aim is to select a subset of covariates $\mathcal{K}_{n,t}$ at each $t$ to maximize the (net) benefit of observing $\{x_{n,t,k}\}_{k \in \mathcal{K}_{n,t}}$.

**As a Software Toolkit**  Engineering *complete* medical time-series workflows is hard. The primary barrier to collaborative research between ML and medicine seldom lies in any particular algorithm. Instead, the difficulty is operational [6, 48, 54]—i.e. in coordinating the entire data science process, from handling missing/irregularly sampled patient data all the way to validation on different popu-

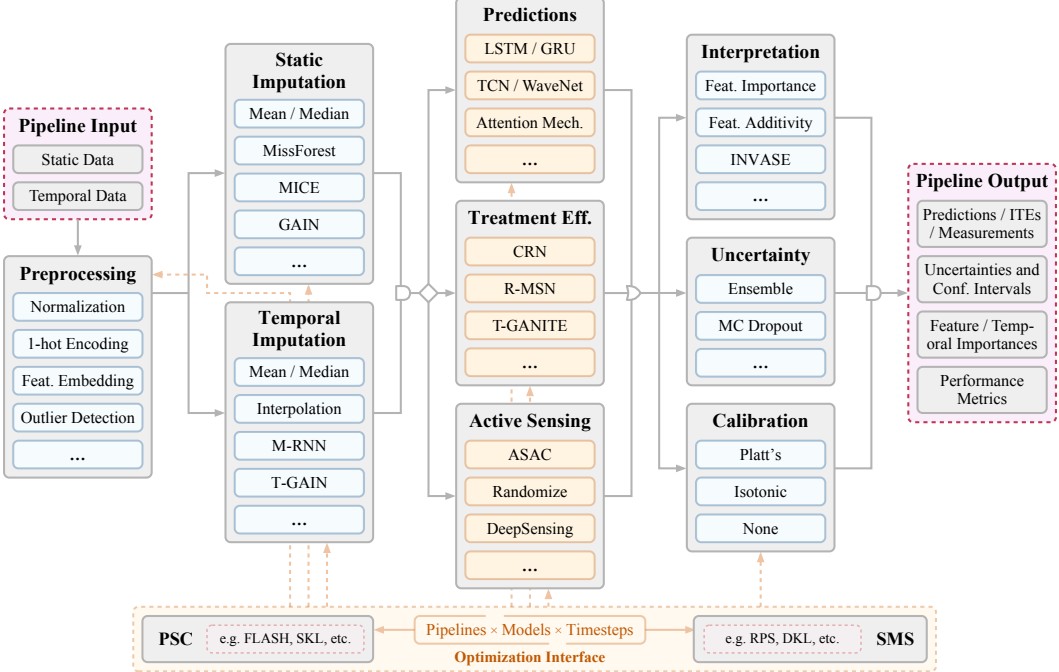

Figure 2: *Clairvoyance Pipeline Overview*. (Dashed) purple cells denote pipeline inputs/outputs, and (solid) gray cells denote pipeline components. Orange options give main pathway models, and blue options give surrounding components. (Solid) gray arrows indicates pipeline workflow, and (dashed) orange the optimization interface.

```
""Configure Data Preprocessing""              ""Configure Pathway Model""
preprocessing = PipelineComposer(              prediction_model = Prediction(model_name='…',
  FilterNegative(…), OneHotEncoder(…),           parameter_dict={…}, …)
  Normalizer(…), …)
                                               ""Load Datasets""
""Configure Problem Specification""            data_train, data_test = DataLoader.load(
specification = ProblemMaker(                     static_dir='…', temporal_dir='…', …),
  problem_class='online', max_seq_len=24,        DataLoader.load(static_dir='…', temporal_dir='…')
  label=['ventilator'], treatment=None, window=4, …)
                                               ""Execute Pipeline""
""Configure Data Imputation""                  for component in [preprocessing,
imputation = PipelineComposer(                                   specification,
  Imputation(type='static', model_name='…', …),                 imputation,
  Imputation(type='temporal', model_name='…', …))               feature_selection]:
                                                 data_train = component.fit_transform(data_train)
""Configure Feature Selection""                  data_test = component.transform(data_test)
feature_selection = PipelineComposer(
  FeatureSelection(type='static', model_name='…', …),  prediction_model.fit(data_train, …)
  FeatureSelection(type='temporal', model_name='…', …))  test_output = prediction_model.predict(data_test, …)
```

Figure 3: *Illustrative Usage*. A prototypical structure of API calls for constructing a prediction pathway model. Clairvoyance is modularized to abide by established fit/transform/predict design patterns. (Green) ellipses denote additional configuration; further modules (treatments, sensing, uncertainty, etc.) expose similar interfaces.

lations [4, 55–60]. Clairvoyance gives a single *unified* roof under which clinicians and researchers alike can readily address such common issues—with the only requirement that the data conform to the standard EAV open schema for clinical records (i.e. patient key, timestamp, parameter, and value).

Under a simple, consistent API, Clairvoyance encapsulates all major steps of time-series modeling, including (a.) loading and (b.) preprocessing patient records, (c.) defining the learning problem, handling missing or irregular samples in both (d.) static and (e.) temporal contexts, (f.) conducting feature selection, (g.) fitting prediction models, performing (h.) calibration and (i.) uncertainty estimation of model outputs, (j.) applying global or instance-wise methods for interpreting learned models, (k.) computing evaluation metrics, and (l.) visualizing results. Figure 2 shows a high-level overview of major components in the pipeline, and Figure 3 shows an illustrative example of usage.

All component modules are designed around the established *fit-transform-predict* paradigms, and the modeling workflow is based around a single chain of API calls. In this manner, each stage in the pipeline is *extensible* with little effort: Novel techniques developed for specific purposes (e.g. a new state-of-the-art imputation method) can be seamlessly integrated via simple wrappers (see Appendix G for an example of how this can be done for any existing method, e.g. from sklearn). This stepwise composability aims to facilitate rapid experimentation and deployment for research, as well as simplifying collaboration and code-sharing. Package documentation/tutorials give further software details.

**As an Empirical Standard** Evaluating any algorithm depends on its *context*. For instance, how well a proposed classifier ultimately performs is invariably coupled with the upstream feature-selection method it is paired with [44]. Likewise, the accuracy of a state-of-the-art imputation method cannot be assessed on its own: With respect to different downstream prediction models, more sophisticated imputation may actually yield inferior performance relative to simpler techniques [13, 14]—especially if components are not jointly optimized [15]. While current research practices typically seek to isolate individual gains through "all-else-equal" configurations in benchmarking experiments, the degree of actual overlap in pipeline configurations *across* studies is lacking: There is often little commonality in the datasets used, preprocessing done, problem types, model classes, and prediction endpoints. This dearth of *empirical standardization* may not optimally promote practical assessment/reproducibility, and may obscure/entangle true progress. (Tables 6–7 in Appendix A give a more detailed illustration).

Clairvoyance aims to serve as a *structured* evaluation framework to provide such an empirical standard. After all, in order to be relevant from a real-world medical standpoint, assessment of any single proposed component (e.g. a novel ICU mortality predictor) can—and should—be contextualized in the entire *end-to-end* workflow as a whole. Together, the 'problem-maker', 'pipeline-composer', and all the pipeline component modules aim to simplify the process of specifying, benchmarking, and (self-)documenting full-fledged experimental setups for each use case. At the end of the day, while results from external validation of is often heterogeneous [2, 59, 61], improving transparency and reproducibility greatly facilitates code re-use and independent verification [54, 56, 57]. Just as the "environment" abstraction in OpenAI Gym does for reinforcement learning, the "pipeline" abstraction in Clairvoyance seeks to promote accessibility and fair comparison as pertains medical time-series.

(a) *Example*: SASH 'decomposed' as SMS (fulfilled here by DKL) followed by combiner (stacking ensemble):

```python
stepwise_pathway_models = []

"Optimize Each Class"
for klass in list_of_pathway_classes:
  sms_agent = Stepwise(method='dkl',
    klass, data_train, metric)
  models, scores = sms_agent.optimize(num_iters=300)

  sms_model = StepwiseEnsemble(models, scores)
  stepwise_pathway_models.append(sms_model)

"Ensemble Over Classes"
for model in stepwise_pathway_models:
  for step in stepwise_model:
    step.load_model(step.get_path(step.model_id))

pathway_model = StackingEnsemble(stepwise_pathway_models)
pathway_model.fit(data_train, …)
test_output = pathway_model.predict(data_test, …)
```

(b) *Example*: SPSC 'decomposed' as PSC (fulfilled here by SKL) followed by SMS (fulfilled here by DKL):

```python
pipeline_classes = [list_of_static_imputation_classes,
  …, …, …, list_of_pathway_classes]

"PSC Optimization"
psc_agent = Componentwise(method='skl',
  pipeline_classes, data_train, data_test, metric)
components, score = psc_agent.optimize(num_iters=300)

pathway_class, data_train, data_test = \
  psc_agent.get_pathway_class_and_data()

"SMS Optimization"
sms_agent = Stepwise(method='dkl',
  pathway_class, data_train, metric)
models, scores = sms_agent.optimize(num_iters=300)

pathway_model = StepwiseEnsemble(models, scores)
pathway_model.load_model(…)
test_output = pathway_model.predict(data_test, …)
```

Figure 4: *Optimization Interface*. Example code using the optimization interface to conduct stepwise (i.e. across time steps) and componentwise (i.e. across the pipeline) configuration. Each interface is implementable by any choice of new/existing algorithms. The DKL implementation of SMS is provided for use the Section 4 examples.

**As an Optimization Interface** Especially in cross-disciplinary clinical research—and during initial stages of experimentation—automated optimization may alleviate potential scarcity of expertise in the specifics of design and tuning. The Clairvoyance pipeline abstraction serves as a software *interface* for optimization algorithms—through which new/existing techniques can be applied, developed, and tested in a more systematic, realistic setting. In particular, by focusing on the temporal aspect of medical time series, this adds a new dimension to classes of autoML problems.

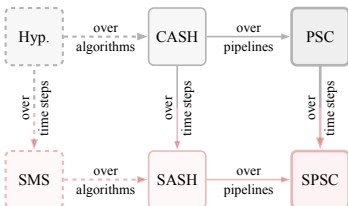

Figure 5: *Degrees of Optimizations*. Clairvoyance allows optimizing over algorithms, pipelines, and time steps.

Briefly (see Figure 5), consider the standard task of hyperparameter optimization (for a given model) [62]. By optimizing over classes of *algorithms*, the combined algorithm selection and hyperparameter optimization ("CASH") problem [63–65] has been approached in healthcare settings by methods such as progressive sampling, filtering, and fine-tuning [50, 66]. By further optimizing over combinations of *pipeline* components, the pipeline selection and configuration ("PSC") problem [67] has also been tackled in clinical modeling via such techniques as fast linear search ("FLASH") [68] and structured kernel learning ("SKL") [67, 69]. Now, what bears further emphasis is that for clinical time series, the *temporal* dimension is critical due to the potential for temporal distribution shifts within time-series data—a common phenomenon in the medical setting (we refer to [53, 70, 71] for additional background). Precisely to account for such temporal settings, the stepwise model selection ("SMS") problem [71] has recently been approached by such methods as relaxed parameter sharing ("RPS") [53] as well as deep kernel learning ("DKL") [71, 72]. Further, what the pipeline interface also does is to naturally allow extending this to define the stepwise algorithm selection and hyperparameter optimization ("SASH") problem, or even—in the most general case—the stepwise pipeline selection and configuration ("SPSC") problem. Although these latter two are new—and clearly hard—problems (with no existing solutions), Figure 4 shows simple examples of how the interface allows minimally adapting the SMS and PSC sub-problems (which do have existing solutions) to form feasible (approximate) solutions.

Two distinctions are due: First, Clairvoyance is a pipeline toolkit, not an autoML toolkit. It is not our goal to (re-)*implement* new/existing optimization algorithms—which abound in literature. Rather, the standardized *interface* is precisely what enables existing implementations to be plugged in, as well as allowing new autoML techniques to be developed and validated within a realistic medical pipeline. All that is required, is for the optimizing agent to expose an appropriate 'optimize' method given candidate components, and for such candidates to expose a 'get-hyperparameter-space' method. Second—but no less importantly—we must emphasize that we are not advocating *removing* human oversight from the healthcare loop. Rather, the pipeline simply encourages *systematizing* the initial development stages in clinical ML, which stands to benefit from existing literature on efficient autoML techniques.

---

**Design Principles**

Our philosophy is based on the authors' experience in prototyping and developing real-world collaborative projects in clinical time-series. • **Pipeline First, Models Second**: Our first emphasis is on *reproducibility*: The process of engineering and evaluating complete medical time-series workflows needs to be clear and transparent. Concretely, this manifests in the strict "separation of concerns" enforced by the high-level API of each component module along the pipeline (see e.g. Figure 3). With the 'problem-maker' and 'problem-composer' as first-class objects, the central abstraction here is the pipeline itself, while the intricacies and configurations of individual model choices (e.g. a specific deep learning temporal imputation method) are limited to within each component module. • **Be Minimal and Unintrusive**: Our second emphasis is on *standardization*: While workflow development needs to be unified and systematic, learning to use the framework should be intuitive as well. Concretely, this manifests in the API's adherence to the existing and popular 'fit-transform-predict' paradigm (see e.g. sklearn) in all component modules—both 'along' the pipeline steps, as well as 'across' the pathways that define the patient's healthcare lifecycle (see Figure 2). This enables easy adoption and rapid prototyping—qualities that are paramount given the degree of collaborative research and cross-disciplinary code-sharing required in healthcare-related research. • **Encourage Extension**: Our third emphasis is on *extensibility*: Given that novel methods are proposed in the ML community every day, the pipeline components should be easily extensible to incorporate new algorithms. Concretely, this manifests in the encapsulated design for models within each component module: Specifically, in order to integrate a new component method (e.g. from another researcher's code, or from an external package) into the framework, all that is required is a simple wrapper class that implements the 'fit', 'predict', and 'get-hyperparameter-space' methods; likewise, for an optimization agent (see subsection on optimization interface below), all that is required is to expose an 'optimize' method.

---

**Worked Examples**

For a discussion of the choice of *built-in* techniques to include with the initial release, see Appendix C. Appendix E gives a worked example of using the Clairvoyance **pipeline** to train and use a model in a standard setting (for this, we use the predictions pathway). Appendix F gives a worked example of using the **optimization** interface to perform stepwise model selection (for this, we use the treatment effects pathway for variety). Appendix G gives an example of how a generic **wrapper** can be written for integrating an external model/algorithm that is not already implemented in the current version of Clairvoyance. Finally, the software repository contains Jupyter notebooks and top-level API code with examples of pathways and optimization.

---

## 3 RELATED WORK

Clairvoyance is a pipeline toolkit for medical time-series machine learning research and clinical decision support. As such, this broad undertaking lies at the intersection of three concurrent domains of work: Time-series software development, healthcare journey modeling, and automated learning.

**Time-series Software** First and foremost, Clairvoyance is a software toolkit. Focusing on challenges common to clinical time-series modeling, it is primarily differentiated by the *breadth* and flexibility of the pipeline. While there exists a variety of sophisticated time-series packages for different purposes, they typically concentrate on implementing collections of algorithms and estimators for specific types of problems, such as classification [79], forecasting [77], feature extraction [76], reductions between tasks [78], or integrating segmentation and transforms with estimators [75]. By contrast, our focus is orthogonal: Clairvoyance aims at end-to-end development along the entire inference workflow, including pathways and pipeline components important to medical problems (see Table 1). Again indeed—if so desired, and as mentioned above—specific algorithms from [73–79] can be integrated into Clairvoyance workflows through the usual 'fit-transform-predict' interface, with little hassle.

**Healthcare Lifecycle** For specific use cases, clearly a plethora of research exists in support of issuing diagnoses [22–24], prognostic modeling [17–21], treatment-effect estimation [26–31], optimizing measurements [32–36], among much more. The key proposition that Clairvoyance advances is the underlying *commonality* across these seemingly disparate problems: It abstracts and integrates along the time-series inference workflow, across outpatient, general wards, and intensive-care environments, and—above all—amongst a patient's journey of interactions through the healthcare system that call for decision support in predictions, treatments, and monitoring (Figure 1). Now, it also important to state what Clairvoyance is *not*: It is not an exhaustive list of algorithms; the pipeline includes a collection of popular components, and provides a standardized interface for extension. It is also not a solution to preference-/application-specific considerations: While issues such as data cleaning, algorithmic fairness, and privacy and heterogeneity are important, they are beyond the scope of our software.

Table 1: *Clairvoyance and Comparable Software.* *Note that vernacularly, "pipelining" simply refers to the *procedural* workflow (i.e. from inputs to training, cross-validation, outputs, and evaluation); existing packages focus on implementing algorithms for prediction models alone, with minimal preprocessing. In contrast, Clairvoyance provides support along the *data science* pipeline, and across different *healthcare pathways* requiring decision support.

| | cesium [73] | tslearn [74] | seglearn [75] | tsfresh [76] | pysf [77] | sktime [78] | pyts [79] | Clairvoyance |
|---|---|---|---|---|---|---|---|---|
| Preprocessing | ✓ | ✓ | ✓ | ✓ | ✗ | ✓ | ✓ | ✓ |
| Temporal Imputation | ✓ | ✓ | ✓ | ✓ | ✓ | ✗ | ✓ | ✓ |
| Feature Selection | ✗ | ✗ | ✗ | ✓ | ✗ | ✗ | ✗ | ✓ |
| *Predictions* Static Features | ✓ | ✗ | ✓ | ✗ | ✗ | ✗ | ✗ | ✓ |
| Online Targets | ✗ | ✗ | ✓ | ✗ | ✓ | ✗ | ✓ | ✓ |
| Predictions | ✓ | ✓ | ✓ | ✓ | ✓ | ✓ | ✓ | ✓ |
| Treatment Effects | ✗ | ✗ | ✗ | ✗ | ✗ | ✗ | ✗ | ✓ |
| Active Sensing | ✗ | ✗ | ✗ | ✗ | ✗ | ✗ | ✗ | ✓ |
| *Interp.* Feat. Importance | ✗ | ✗ | ✗ | ✗ | ✗ | ✗ | ✗ | ✓ |
| Feat. Additivity | ✗ | ✗ | ✗ | ✗ | ✗ | ✗ | ✗ | ✓ |
| Instance-wise | ✗ | ✗ | ✗ | ✗ | ✗ | ✗ | ✗ | ✓ |
| Uncertainty | ✗ | ✗ | ✗ | ✗ | ✗ | ✗ | ✗ | ✓ |
| Calibration | ✗ | ✗ | ✗ | ✗ | ✗ | ✗ | ✗ | ✓ |
| End-to-End Pipelining* | ✗ | ✗ | ✗ | ✗ | ✗ | ✗ | ✗ | ✓ |
| Optimization Interface | ✗ | ✗ | ✗ | ✗ | ✗ | ✗ | ✗ | ✓ |

**Automated Learning** Finally, tangentially related is the rich body of work on autoML for hyperparameter optimization [62], algorithm/pipeline configuration [63–65, 67], and stepwise selection [71], as well as specific work for healthcare data [50, 53, 66–68, 70, 71]. In complement to these threads of research, the Clairvoyance pipeline interface enables—if so desired—leveraging existing implementations, or validating novel ones—esp. in efficiently accounting for the temporal dimension.

## 4 ILLUSTRATIVE EXAMPLES

Recall the patient's journey of interactions within the healthcare system (Figure 1). In this section, our goal is to illustrate *key usage scenarios* for Clairvoyance in this journey—for personalized (1) prediction, (2) treatment, and (3) monitoring—in outpatient, general wards, and intensive-care environments.

Specifically, implicit in all examples is our proposition that: (i) as a software toolkit, constructing an end-to-end solution to each problem is *easy*, *systematic*, and *self-documenting*; (ii) as an empirical standard, evaluating collections of models by varying a single component ensures that comparisons are *standardized*, *explicit*, and *reproducible*; and (iii) as an optimization interface, the flexibility of selecting over the *temporal dimension*—in and of itself—abstracts out an interesting research avenue.

**Medical Environments** Our choices of time-series environments are made to reflect the heterogeneity of realistic use cases envisioned for Clairvoyance. For the outpatient setting, we consider a cohort of patients enrolled in the UK Cystic Fibrosis Registry (**CYSTIC**) [80], which records longitudinal follow-up data for ∼5,800 individuals with the disease. On the registry, individuals

| *Medical Environment* | **Outpatient** | **General Wards** | **Intensive Care** |
|---|---|---|---|
| Dataset | UKCF [80] | WARDS [81] | MIMIC [82] |
| Duration of Trajectories | Avg. ∼5.3 years | Avg. ∼9.1 days | Avg. ∼85.4 hours |
| Variance (25%–50%–75%) | (4–6–7 years) | (6–9–15 days) | (27–47–91 hours) |
| Frequency of Measurements | Per 6 months | Per 4 hours | Per 1 hour |
| Different Types of Static | Demo., Comorbidities, | Admiss. Stats, Vital | Demo., Vital Signs, |
| and Temporal Features | Infections, Treatments | Signs, Lab Tests | Lab Tests, Medications |
| Dimensionality of Features | 11 static, 79 temporal | 8 static, 37 temporal | 11 static, 40 temporal |
| Number of Samples | ∼5,800 patients | ∼6,300 patients | ∼23,100 patients |
| Endpoints (cf. Predictions) | FEV1 Result | Admission to ICU | Mechanical Ventilation |
| Class-label Imbalance | (Continuous-valued) | ∼5.0%-to-95.0% | ∼36.8%-to-63.2% |

Table 2: *Medical Environments.* We consider the range of settings, incl. outpatient, general wards, and ICU data.

are *chronic patients* monitored over infrequent visits, and for which long-term decline is generally expected. For the general wards setting, we consider a cohort of ~6,300 patients hospitalized in the general medicine floor in the Ronald Reagan Medical Center (**WARDS**) [81]. In contrast, here the population of patients presents with a wide variety of conditions and diagnoses (1,600+ ICD-9 codes), and patients are monitored more frequently. The data is highly non-stationary: on the hospital floor, deterioration is an *unexpected* event. For the intensive-care setting, we consider ~23,100 individuals from the Medical Information Mart for Intensive Care (**MIMIC**) [82]. Here, the setting is virtually that more or less "anything-can-happen", and physiological data streams for each patient are recorded extremely frequently. Varying across the set of environments are such characteristics as the average durations of patient trajectories, the types of static and longitudinal features recorded, their frequencies of measurement, and their patterns and rates of missingness (Table **??** presents some brief statistics).

**Example 1 (Lung Function in Cystic Fibrosis Patients)** The most common genetic disease in Caucasian populations is cystic fibrosis [83], which entails various forms of dysfunction in respiratory and gastrointestinal systems, chiefly resulting in progressive lung damage and recurrent respiratory infections requiring antibiotics—and in severe cases may require hospitalization and even mechanical ventilation in an ICU (see Example 2) [84, 85]. While classical risk scores and survival models utilize only a fraction of up-to-date measurements, recent work has leveraged deep learning to incorporate greater extents of longitudinal biomarkers, comorbidities, and other risk factors [86]. An essential barometer for anticipating the occurrence of respiratory failures is the gauge of lung function by *forced expiratory volume* (FEV1): Accurate prediction yields an important tool for assessing severity of a patient's disease, describing its onset/progression, and as an input to treatment decisions [85, 87].

This is an archetypical rolling-window time-series problem for Clairvoyance's *predictions pathway*. Consider the models in Table 3: (i) As a clinical professional, it goes without saying that building the pipeline for each—or extending additional models through wrappers—has a low barrier to entry (see Figure 3/tutorials/documentation). (ii) As an ML researcher, one can rest assured that such comparisons are expressly standardized: Here, all results are explicitly from same pipeline using min-max normalized features, GAIN for static missing values, M-RNN for temporal imputation, no feature selection, and each model class shown. (iii) Lastly, to highlight the utility of the interface for selection over time, the final row presents results of approaching SASH using the example method of Figure 4(a), and—for fair comparison—with the pipeline kept constant. This simple approach already yields some gains in performance, laying a precedent—and the pipeline infrastructure—for further research.

| *Dataset* (*Label*) | UKCF (FEV1 Result) | | WARDS (Admission to ICU) | | MIMIC (Mech. Ventilation) | |
|---|---|---|---|---|---|---|
| Evaluation | RMSE | MAE | AUC | APR | AUC | APR |
| Attention | (N/A) | (N/A) | 0.888 ± 0.016 | 0.551 ± 0.024 | (N/A) | (N/A) |
| RNN-GRU | 0.064 ± 0.001 | 0.035 ± 0.001 | 0.865 ± 0.010 | 0.487 ± 0.048 | 0.898 ± 0.001 | 0.774 ± 0.002 |
| RNN-LSTM | 0.062 ± 0.001 | 0.033 ± 0.001 | 0.841 ± 0.014 | 0.412 ± 0.032 | 0.901 ± 0.001 | 0.776 ± 0.002 |
| Temporal CNN | 0.120 ± 0.004 | 0.096 ± 0.003 | 0.826 ± 0.020 | 0.319 ± 0.048 | 0.884 ± 0.004 | 0.749 ± 0.007 |
| Transformer | 0.081 ± 0.002 | 0.050 ± 0.002 | 0.846 ± 0.006 | 0.472 ± 0.045 | 0.889 ± 0.002 | 0.761 ± 0.004 |
| Vanilla RNN | 0.070 ± 0.001 | 0.043 ± 0.001 | 0.794 ± 0.018 | 0.277 ± 0.063 | 0.898 ± 0.001 | 0.771 ± 0.002 |
| **SASH** | **0.059 ± 0.001** | **0.030 ± 0.001** | **0.891 ± 0.011** | **0.557 ± 0.031** | **0.917 ± 0.006** | **0.809 ± 0.013** |

Table 3: *Predictions Pathway Example*. In addition to (online) 6-month ahead predictions of FEV1 in UKCF, we also test (one-shot) predictions of admission to ICU after 48 hours on the floor in WARDS, and (online) 4-hours ahead predictions of the need for mechanical ventilation in MIMIC (these are extended to treatment and sensing problems below). As the WARDS prediction is one-shot, what is denoted 'SASH' for that excludes the SMS ensembling step. Note that the canonical attention mechanism does not permit (variable-length) online predictions.

**Example 2 (Mechanical Ventilation on Intensive Care)** Mechanical ventilation is an invasive, painful, and extremely unpleasant therapy that requires induction of artificial coma, and carries a high risk of mortality [88]. It is also expensive, with a typical ICU ventilator admission >$30,000 [89]. To the patient, the need for mechanical ventilation—due to evidence of respiratory/ventilatory failure—is by itself an adverse outcome, and is unacceptable to some, even if it means they will not survive. It is possible that alternative strategies employed earlier may alleviate the need for ventilation, such as high flow oxygen, non-invasive ventilation, or—in this example—appropriate *use of antibiotics* [88]. Now, little is known about optimal timing of courses of antibiotics; in most cases a routine number of days is simply chosen when blood is typically sterile after first dose. On the one hand, there is a clear biologically plausible mechanism for incompletely treated infection to lead to longer periods of

critical care, esp. requiring ventilation. On the other hand, antibiotic stewardship is crucial: Over-use of broad spectrum antibiotics leads to resistance, and is by itself a global health emergency [90].

This is an archetypical problem for the *treatment effects pathway*. Table 4 shows the performance of the two state-of-the-art models for estimating effects of treatment decisions over time while adjusting for time-dependent confounding—that is, since actions taken in the data may depend on time-varying variables related to the outcome of interest [30, 31]. We refrain from belaboring points (i), (ii), (iii) above but their merits should be clear. From the *patient*'s perspective, accurate estimation of the effect of treatment decisions on the risk of ventilation may assist them and their carers in achieving optimal shared decision-making about the care that they would like to receive. From the *hospital*'s perspective, many ICUs around the world operate at ∼100% bed occupancy, and delayed admission is typically an independent predictor of mortality [91–94]; therefore accurate estimation of the need for escalation or continued ICU ventilation is logistically important for resource planning and minimization of delays.

| *Time Horizon* | Estimating 1 Day Ahead | | Estimating 2 Days Ahead | | Estimating 3 Days Ahead | |
|---|---|---|---|---|---|---|
| Evaluation | AUC | APR | AUC | APR | AUC | APR |
| RMSN | $0.860 \pm 0.005$ | $0.889 \pm 0.007$ | $0.790 \pm 0.004$ | $0.883 \pm 0.003$ | $0.726 \pm 0.015$ | $0.852 \pm 0.009$ |
| CRN | $0.865 \pm 0.003$ | $0.892 \pm 0.004$ | $0.783 \pm 0.009$ | $0.872 \pm 0.013$ | $0.767 \pm 0.010$ | $0.869 \pm 0.007$ |
| SASH | $\mathbf{0.871 \pm 0.007}$ | $\mathbf{0.902 \pm 0.005}$ | $\mathbf{0.792 \pm 0.003}$ | $\mathbf{0.885 \pm 0.009}$ | $\mathbf{0.771 \pm 0.005}$ | $\mathbf{0.873 \pm 0.003}$ |

Table 4: *Treatment Effects Pathway Example*. Results for estimation over different horizon lengths. Note that this uses a ∼6,000-patient subset (from those in Table **??**) who received antibiotics at any point, based on daily decisions on antibiotic treatment, over spans of up to 20 days, with labels distributed 58.9%-to-41.1% overall.

**Example 3 (Clinical Deterioration of Ward Patients)** Given the delay-critical nature of ICU admission w.r.t. morbidity/mortality, what is often desired is an automated prognostic decision support system to monitor ward patients and raise (early) alarms for impending admission to ICU (as a result of clinical deterioration) [25, 94, 95]. However, observations are costly, and the question of what (and when) to measure is by itself an *active choice* under resource constraints [32–36]: For instance, there is less reason to measure a feature whose value can already be confidently estimated on the basis of known quantities, or if its value is not expected to contribute greatly to the prognostic task at hand.

This is an archetypical problem for Clairvoyance's *active sensing pathway*. Table 5 indicates the performance of different models for balancing this trade-off between information gain and acquisition rate with respect to admissions to ICU of ward patients. At various budget constraints (i.e. amounts of measurements permitted), each active sensing model learns from the training data to identify the most informative features to measure at test-time, so as to maximize the performance of admission predictions. (To allow some measurements to be costlier than others, they can simply be up-weighted when computing the budget constraint). As before, our propositions (i), (ii), and (iii) are implicit here.

| *Measure Rate* | With 50% Measurements | | With 70% Measurements | | With 90% Measurements | |
|---|---|---|---|---|---|---|
| Evaluation | AUC | APR | AUC | APR | AUC | APR |
| ASAC | $0.714 \pm 0.018$ | $0.235 \pm 0.034$ | $0.781 \pm 0.015$ | $0.262 \pm 0.037$ | $0.841 \pm 0.016$ | $0.414 \pm 0.033$ |
| DeepSensing | $0.707 \pm 0.020$ | $0.230 \pm 0.036$ | $0.772 \pm 0.016$ | $0.255 \pm 0.033$ | $0.829 \pm 0.017$ | $0.409 \pm 0.038$ |
| Randomize | $0.677 \pm 0.021$ | $0.217 \pm 0.033$ | $0.729 \pm 0.019$ | $0.249 \pm 0.032$ | $0.788 \pm 0.017$ | $0.269 \pm 0.039$ |
| SASH | $\mathbf{0.725 \pm 0.015}$ | $\mathbf{0.248 \pm 0.032}$ | $\mathbf{0.793 \pm 0.013}$ | $\mathbf{0.278 \pm 0.043}$ | $\mathbf{0.849 \pm 0.014}$ | $\mathbf{0.420 \pm 0.037}$ |

Table 5: *Active Sensing Pathway Example*. Results at different acquisition rates (using GRUs as base predictors).

## 5 CONCLUSION

Machines will never replace a doctor's medical judgment, nor an ML researcher's technical innovation. But as a matter of data-driven *clinical decision support*, Clairvoyance enables rapid prototyping, benchmarking, and validation of complex time-series pipelines—so doctors can spend more time on the real scientific problems, and ML researchers can focus on the real technical questions. Moreover, collaborative research between medical practitioners and ML researchers is increasingly common [48]. To help grease the wheels, we developed and presented Clairvoyance, and illustrated its flexibility and capability in answering important and interesting medical questions in real-world environments.

ACKNOWLEDGMENTS

We would like to thank the reviewers for their generous and invaluable comments and suggestions. This work was supported by Alzheimer's Research UK (ARUK), The Alan Turing Institute (ATI) under the EPSRC grant EP/N510129/1, The US Office of Naval Research (ONR), and the National Science Foundation (NSF) under grant numbers 1407712, 1462245, 1524417, 1533983, and 1722516.

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

## A    NEED FOR EMPIRICAL STANDARDIZATION

*'All else' is seldom 'equal'.* As examples, a review of recent, state-of-the-art research on medical time-series imputation and prediction models demonstrates the following: While the benchmarking performed *within* each individual study strives to isolate sources of gain through "all-else-equal" experiments, the degree of overlap in pipeline settings *across* studies is lacking. Such a dearth of empirical standardization may not optimally promote effective assessment of true research progress:

| Proposed Imputation Method | | Downstream Prediction Component | | | | Dataset(s) Used |
|---|---|---|---|---|---|---|
| Technique | Evaluation | Problem Type | Endpoint | Model(s) | Evaluation | |
| med.impute [12] | Imputation MSE | One-shot Classification | 10-year Risk of Stroke | LogR | Prediction ROC | Framingham Heart St. (FHS) |
| BRITS [13] | Imputation MAE, MRE | One-shot Classification | In-Hospital Death | NN | Prediction ROC | PhysioNet ICU (MIMIC) |
| GRUI-GAN [14] | None | One-shot Classification | In-Hospital Death | LogR, SVM, RF, RNN | Prediction ROC | PhysioNet ICU (MIMIC) |
| GP-based [16] | Imputation MSE | None | N/A | N/A | N/A | UF Shands Hospital Data |
| M-RNN [15] | Imputation MSE | Online Classification | Various Endpoints[1] | NN, RF, LogR, XGB | Prediction ROC | Various Medical Datasets[2] |

Table 6: *Research on Medical Time-series Data Imputation.* Typically, proposed imputation methods rely on some downstream dummy prediction task for evaluating utility. However, the datasets, problem types, prediction endpoints, and time-series models themselves do not often coincide. Depending on specific use cases, this dearth of standardized benchmarking does not promote accessible comparison across different proposed techniques.

| Proposed Prediction Method | | Upstream Imputation Component | | | | Dataset(s) Used |
|---|---|---|---|---|---|---|
| Technique | Endpoint | Evaluation | Imputation | Model(s) | Evaluation | |
| LSTM-DO-TR [22] | Multi-label Diagnosis | ROC, F1, precision at 10 | Yes | Forw-, Back-, Mean-fill | None | Ch. Hospital LA PICU |
| T-LSTM [23] | Regression; Subtyping | MSC; tests for group effects | Data Pre-imputed | N/A | None | Parkinson's (PPMI) |
| MGP-RNN [37] | Sepsis Onset Prediction | ROC, PRC, precision | Yes | Multitask GPs | None | Duke UHS Inpatient |
| D-Atlas [19] | Survival; Forecasting | ROC, PRC, MSE | Yes | Median-, Mean-fill | None | Cystic Fibrosis (UKCF) |
| SAND [24] | Regression; Classification | ROC, PRC, MSE, MAPE | Masking as Input | N/A | None | PhysioNet ICU (MIMIC) |

Table 7: *Research on Medical Time-series Prediction Models.* Typically, proposals of prediction models pay short attention to the choices regarding upstream imputation of missing and/or irregularly sampled data. In comparative experiments, a single imputation method is usually fixed w.r.t. all prediction models for evaluation. The datasets, imputation methods, and even prediction endpoints themselves have little overlap across studies.

# B    BACKGROUND REFERENCES FOR EXPERIMENTS

|  | Example 1 | Example 2 | Example 3 |
|---|---|---|---|
| **Modeling Question** | Lung Function in Cystic Fibrosis Patients | Mechanical Ventilation on Intensive Care | Clinical Deterioration of Ward Patients |
| **Original description of dataset** | Data Resource Profile: The UK Cystic Fibrosis Registry [80] | The Medical Information Mart for Intensive Care [82] | The Ronald Reagan Medical Center General Wards Dataset [81] |
| **Initial data cleaning/ selection used** | As specified in the original study in [86], specifically as in their Appendix A | As specified in the software corresponding to the implementation of the study in [96] | As specified in the original study in [34], specifically as in their Section 5 |
| **Theoretical background on model pathway involved** | This is vanilla time-series prediction, so any technical study should have adequate background, such as our references [17] through [25] in the introduction of this paper | Decision support with counterfactuals [28], marginal structural models [26] and their deep equivalents [30], and the most recent: counterfactual recurrent networks [31] | Active sensing as an original problem [32], models for personalized screening [33], and most recent deep learning methods for the active sensing problem [34–36] |
| **Original works describing and justifying the modeling task** | Background on Cystic Fibrosis [83, 84], studies of classical risk scores [85], studies of joint modeling approaches for survival analysis [87], as well as more recent deep learning approaches to survival analysis [86] | Mechanical ventilation as an adverse outcome of interest [88, 89], ventilation as pertains mortality [91–94], and the importance of antibiotic stewardship vs. reduction in the need for critical care [90] | Prior methods and importance of forecasting deterioration of patients in general wards [25, 95], and specifically developing forecasting systems for for clinical decision support [94] |
| **Data collection, data coverage, ethics, etc.** | See Sections 1–3 in [80] | See Sections 2–3 in [82] | See Section 4 in [81] |

Table 8: *Background References*. Additional table of background references for the experiments in Section 4.

# C    NOTE ON CHOICE OF BUILT-IN TECHNIQUES

Given our "Pipeline First" focus (see "Key Design Principles" in Section 2), especially in the context of medical applications, rather than (re-)implementing every time-series model in existence, our primary contribution is in unifying all three key pathways in a patient's healthcare lifecycle (i.e. predictions, treatments, and monitoring tasks; see Section 2: "The Patient Journey") through a single end-to-end pipeline abstraction—for which Clairvoyance is the first (see Table 1: "Clairvoyance and Comparable Software"). For the predictions pathway, while there is a virtually infinite variety of time-series models in the wild, we choose to include standard and popular classes of *deep learning* models, given their ability to handle large amounts and dimensions of data, as well as the explosion of their usage in medical time-series studies (see e.g. virtually any of the paper references in Section 1). For both the treatment effects and active sensing pathways, there is much less existing work available; for these, we provide state-of-the-art models (e.g. CRN, R-MSN, ASAC, DeepSensing) implemented exactly as given in their original research papers. With that said, as noted throughout, recall that all component modules (including the various other pipeline components) are easily *extensible*: For instance, if more traditional time-series baselines from classical literature were desired for comparison purposes, existing algorithms from packages such as [73–79] can be integrated into Clairvoyance by using simple wrapper classes with little hassle (for an explicit demonstration of this, see Appendix G).

**Note on Time to Train**: Our computations for the examples included in Section 4 were performed using a single NVIDIA GeForce GTX 1080 Ti GPU, and each experiment took approximately ~24–72 hours. Of course, this duration may be shortened through the use of multiple GPUs in parallel.

# D   ADDITIONAL DETAIL ON EXPERIMENTS

In our experiments for UKCF (used in Example 1), out of the total of 10,995 entries in the registry data, we focused on the 5,883 adult patients with followup data available from January 2009 through December 2015, which excludes pediatric patients and patients with no follow-up data from January 2009. This includes a total of 90 features, with 11 static covariates and 79 time-varying covariates, which includes basic demographic features, genetic mutations, lung function scores, hospitalizations, bacterial lung infections, comorbidities, and therapeutic management. Within the 5,883 patients, 605 were followed until death (the most common causes of which were complications due to transplantation and CF-associated liver disease); the remaining 5,278 patients were right-censored.

In our experiments for WARDS (used in Examples 1 and 3), the data comes from 6,321 patients who were hospitalized in the general medicine floor during the period March 2013 through February 2016, and excludes patients who were reverse transfers from the ICU (i.e. initially admitted from the ICU, and then returned to the ward subsequent to stabilization in condition). The heterogeneity in patient conditions mentioned in the main text include such conditions as shortness of breath, hypertension, septicemia, sepsis, fever, pneumonia, and renal failure. Many patients had diagnoses of leukemia or lymphoma, and had received chemotherapy, allogeneic or autologous stem cell transplantation, and treatments that cause severe immunosuppression places them at risk at developing further complications that may require ICU admission. Here, the recorded features include 8 static variables (admission-time statistics) and 37 temporal physiological data streams (vital signs and laboratory tests); vital signs were taken approximately every 4 hours, and lab tests approximately every 24 hours.

In our experiments for MIMIC (used in Example 1 for predictions, and Example 2 for estimating treatment effects), for the predictions example we focus on 22,803 patients who were admitted to ICU after 2008, and consider 11 static variables (demographics information) and 40 physiological data streams in total, which includes 20 vital signs which were most frequently measured and for which missing rates were lowest (e.g. heart rate, respiratory rate), as well as 20 laboratory tests (e.g. creatinine, chloride); vital signs were taken approximately every 1 hour, and laboratory tests approximately every 24 hours. For the treatment effects pathway (used in Example 2), we focus on the 6,033 patients who had received antibiotics at any point in time, based on daily decisions on antibiotic treatment, with a maximum sequence length of 20 days. Note that the class-label imbalance between the pure prediction task (Example 1) and treatment effects task (Example 2) is slightly different per the different populations included, and the numerical results should not be compared directly. The code for extracting this data is included under 'mimic_data_extraction' in the repository.

In all experiments, the entire dataset is first randomly partitioned into training sets (64%), validation sets (16%), and testing sets (20%). The training set is used for model training, the validation set is used for hyperparameter tuning, and the testing set is used for the final evaluation—which generates the performance metrics. This process itself is then repeated randomly for a total of 10 times, with the means and spreads of each result used in generating results Tables 3–5. As usual, the entire pipeline (with the exception of the pathway model corresponding to each row) is fixed across all rows, which in this case uses min-max normalized features, GAIN for static missing values, M-RNN for temporal imputation, and no prior feature selection; where hyperparameters for such pipeline components are involved (i.e. GAIN and M-RNN here), these are also—as they should be—constant across all rows.

In order to highlight our emphasis on the temporal dimension of autoML in Clairvoyance, the results for SASH isolate precisely this effect alone: Each result for SASH is generated using the simple approach of Figure 4(a)—that is, by 'naively' decomposing SASH into a collection of SMS problems (for each model class considered), subsequent to which the stepwise models for each class are further ensembled through stacking. Note that the point here is not to argue for this specific technique, but merely to show that even this (simplistic) approach already yields some gains, thereby illustrating the potential for further autoML research (which can be conveniently performed over Clairvoyance's pipeline abstraction) to investigate perhaps more efficient solutions with respect to this temporal dimension. Briefly, in DKL the validation performance for each time step is treated as a noisy version of a black box function, which leads to a multiple black-box function optimization problem (which DKL solves jointly and efficiently); we refer to [71] for their original exposition. In our experiments we complete 100 iterations of Bayesian optimization in DKL for each model class. For reproducibility, the code for our implementation of DKL used for experiments is included in the repository.

# E WORKED EXAMPLE: USING THE FULL PIPELINE

This section gives a fully worked example of using the Clairvoyance pipeline (via the predictions pathway). To follow along, the user should have their own static and temporal datasets for training and testing, named as follows—where 'data_name' is replaced by the appropriate name of the dataset:

- data_name_temporal_train_data_eav.csv.gz
- data_name_static_train_data.csv.gz
- data_name_temporal_test_data_eav.csv.gz
- data_name_static_test_data.csv.gz

and placed within the directory '../datasets/data/data_name/'. As described in Section 2 ("As a Software Toolkit"), the requirement is that the data conform to the standard EAV open schema for clinical records (i.e. patient key, timestamp, parameter, and value). See Figure 6 for a summary of the pipeline workflow that we shall be walking through and executing in the following subsections:

1. *Load Dataset*: Extract csv files from the original raw datasets located in the data directory.
2. *Preprocess Dataset*: Preprocess the raw data using various filters, such as replacing negative values to NaN, doing one-hot encoding for certain features, and normalizing feature values.
3. *Define Problem*: Set the prediction problem (one-shot or online), the label (the target of predictions), the maximum sequence length, and (optionally) the treatment features (not used here). Also define the metric for evaluation and the task itself (classification or regression).
4. *Impute Dataset*: Impute missing values in the preprocessed static and temporal datasets—for each selecting among data imputation methods of choice, and return the complete datasets.
5. *Feature Selection*: Select the relevant static and temporal features for the labels (e.g. recursive or greedy addition/deletion, or simply skip this step by setting the method to be None.
6. *Model Training and Prediction*: After finishing the data preparation steps, we define the model used for time-series prediction, and train the model using the training dataset. After training is finished, we use the trained model to predict the labels using the testing dataset.
7. *Estimate Uncertainty*: Estimate uncertainty of the predictions made by the predictor model.
8. *Interpret Predictions*: Compute the (instance-wise) feature and temporal importance weights.
9. *Visualize Results*: Output predictions, performance metrics, uncertainties, and importances.

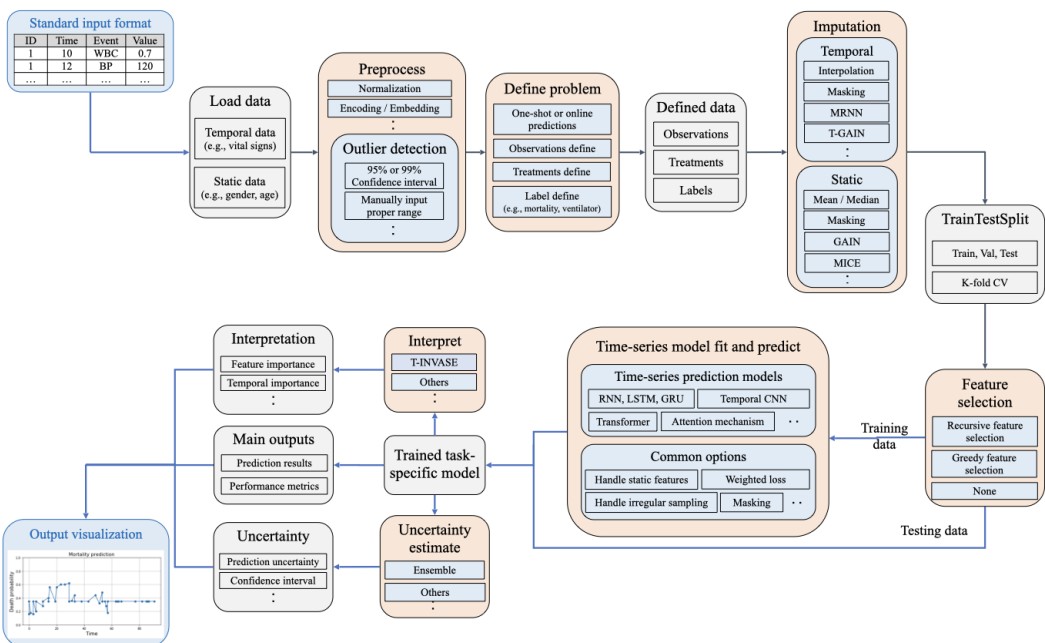

Figure 6: *Pipeline Workflow*. Step-by-step schematic corresponding to the procedure in this worked example.

Import necessary packages for this example:

```
# Necessary packages
from __future__ import absolute_import
from __future__ import division
from __future__ import print_function

import numpy as np
import warnings; warnings.filterwarnings('ignore')
import sys; sys.path.append('../')

from utils import PipelineComposer
```

## E.1 LOAD DATASET

Extract `csv` files from the original raw datasets located in the directory. The `CSVLoader` is responsible for loading `csv` files from the original raw datasets in the '`../datasets/data/data_name/`' directory. In this example we use data from MIMC, so here the '`data_name`' is '`mimic`' throughout:

```
───── Load Dataset ─────
from datasets import CSVLoader

# Define data name
data_name = 'mimic'

# Define data directory
data_directory = '../datasets/data/'+data_name + '/' + data_name + '_'

# Load train and test datasets
data_loader_training = \
  CSVLoader(static_file=data_directory + 'static_train_data.csv.gz',
            temporal_file=data_directory + 'temporal_train_data_eav.csv.gz')

data_loader_testing = \
  CSVLoader(static_file=data_directory + 'static_test_data.csv.gz',
            temporal_file=data_directory + 'temporal_test_data_eav.csv.gz')

dataset_training = data_loader_training.load()
dataset_testing = data_loader_testing.load()

print('Finish data loading.')
```

## E.2 PREPROCESS DATASET

Preprocess the raw data using multiple filters. In this example, we replace all the negative values to NaN (using `FilterNegative`), do one-hot encoding on '`admission_type`' feature (using `OneHotEncoder`), and do MinMax Normalization (using `Normalizer`). Preprocessing is done for both training and testing datasets; note that—as should be the case—the '`fit_transform`' method is called on the training dataset, and only the '`transform`' method is executed on the testing dataset:

```
───── Preprocess Dataset ─────
from preprocessing import FilterNegative, OneHotEncoder, Normalizer

# (1) filter out negative values
negative_filter = FilterNegative()

# (2) one-hot encode categorical features
one_hot_encoding = 'admission_type'
onehot_encoder = OneHotEncoder(one_hot_encoding_features=[one_hot_encoding])

# (3) Normalize features: 3 options (minmax, standard, none)
```

```
normalization = 'minmax'
normalizer = Normalizer(normalization)

# Data preprocessing
filter_pipeline = PipelineComposer(negative_filter, onehot_encoder, normalizer)

dataset_training = filter_pipeline.fit_transform(dataset_training)
dataset_testing = filter_pipeline.transform(dataset_testing)

print('Finish preprocessing.')
```

### E.3 DEFINE PROBLEM

The prediction problem can be defined as: 'one-shot' (one time prediction) or 'online'(rolling window prediction). The "max_seq_len' is the maximum sequence length of time-series sequence. The 'label_name' is the column name for the label(s) selected as the prediction target. The 'treatment' is the column name for the actions selected as treatments (not used here in this example, in the predictions pathway). The 'window' specifies the prediction window (i.e. how many hours ahead to predict). The 'metric_name' specifies the performance metric of interest, e.g. 'auc', 'apr', 'mse', 'mae', and the 'task' is classification or regression. In this example, we are interested in issuing online predictions for whether the patient will require mechanical ventilation after 4 hours:

```
────────── Define Problem ──────────
from preprocessing import ProblemMaker

# Define parameters
problem = 'online'
max_seq_len = 24
label_name = 'ventilator'
treatment = None
window = 4

# Define problem
problem_maker = \
  ProblemMaker(problem=problem, label=[label_name],
               max_seq_len=max_seq_len, treatment=treatment, window = window)

dataset_training = problem_maker.fit_transform(dataset_training)
dataset_testing = problem_maker.fit_transform(dataset_testing)

# Set other parameters
metric_name = 'auc'
task = 'classification'

metric_sets = [metric_name]
metric_parameters =  {'problem': problem, 'label_name': [label_name]}

print('Finish defining problem.')
```

### E.4 IMPUTE DATASET

For static imputation there are options such as mean, median, mice, missforest, knn, gain. For temporal imputation there are options such as mean, median, linear, quadratic, cubic, spline, mrnn, tgain. In this example we simply select median imputation for both static and temporal data:

```
────────── Impute Dataset ──────────
from imputation import Imputation

# Set imputation models
static_imputation_model = 'median'
temporal_imputation_model = 'median'
```

```
# Impute the missing data
static_imputation = Imputation(imputation_model_name = static_imputation_model,
                               data_type = 'static')
temporal_imputation = Imputation(imputation_model_name = temporal_imputation_model,
                                 data_type = 'temporal')

imputation_pipeline = PipelineComposer(static_imputation, temporal_imputation)

dataset_training = imputation_pipeline.fit_transform(dataset_training)
dataset_testing = imputation_pipeline.transform(dataset_testing)

print('Finish imputation.')
```

## E.5 FEATURE SELECTION

In this step, we can perform feature selection for the most relevant static and temporal features to the labels. In the simplest case, we can skip the feature selection step entirely (as we do here). The user can select from among `greedy-addtion`, `greedy-deletion`, `recursive-addition`, `recursive-deletion`, and `None`. The `feature_number` specifies the number of selected features:

```
──────── Feature Selection ────────
from feature_selection import FeatureSelection

# Set feature selection parameters
static_feature_selection_model = None
temporal_feature_selection_model = None
static_feature_selection_number = None
temporal_feature_selection_number = None

# Select relevant features
static_feature_selection = \
FeatureSelection(feature_selection_model_name = static_feature_selection_model,
                 feature_type = 'static',
                 feature_number = static_feature_selection_number,
                 task = task, metric_name = metric_name,
                 metric_parameters = metric_parameters)

temporal_feature_selection = \
  FeatureSelection(feature_selection_model_name = temporal_feature_selection_model,
                 feature_type = 'temporal',
                 feature_number = temporal_feature_selection_number,
                 task = task, metric_name = metric_name,
                 metric_parameters = metric_parameters)

feature_selection_pipeline = \
  PipelineComposer(static_feature_selection, temporal_feature_selection)

dataset_training = feature_selection_pipeline.fit_transform(dataset_training)
dataset_testing = feature_selection_pipeline.transform(dataset_testing)

print('Finish feature selection.')
```

## E.6 TRAINING AND PREDICTION

After finishing the data preparation, we define the predictive models. Existing options include RNN, GRU, LSTM, Attention, Temporal CNN, and Transformer, and—as is the case for the other pipeline modules, and as discussed in Section 2—is easily extensible through the standard *fit-transform-predict* paradigm. We now train the model using the training dataset. We set the validation set as the 20% of the training set for early stopping and for saving the best model. After training, we use the trained model to predict the labels of the testing dataset. Here the parameters include `model_name`: rnn, gru, lstm, attention, tcn, transformer; `model_parameters`: network parameters,

such as `hdim`: hidden dimensions, `n_layer`: number of layers, `n_head`: number of heads (for transformer model), `batch_size`: number of samples in mini-batch, `epochs`: number of epochs, `learning_rate`: learning rate, `static_mode`: method of incorporating static features (e.g. by concatenation), `time_mode`: method of incorporating temporal information (e.g. concatenate), etc.:

```
──────────────── Training and Prediction ────────────────
from prediction import prediction

# Set predictive model
model_name = 'gru'

# Set model parameters
model_parameters = {'h_dim': 100,
                    'n_layer': 2,
                    'n_head': 2,
                    'batch_size': 128,
                    'epoch': 20,
                    'model_type': model_name,
                    'learning_rate': 0.001,
                    'static_mode': 'Concatenate',
                    'time_mode': 'Concatenate',
                    'verbose': True}

# Set up validation for early stopping and best model saving
dataset_training.train_val_test_split(prob_val=0.2, prob_test = 0.0)

# Train the predictive model
pred_class = prediction(model_name, model_parameters, task)
pred_class.fit(dataset_training)

# Return the predictions on the testing set
test_y_hat = pred_class.predict(dataset_testing)

print('Finish predictor model training and testing.')
```

### E.7 ESTIMATE UNCERTAINTY

Estimate uncertainty of the predictions (which we name '`test_ci_hat`' below) made by the predictor model. In this example, we use the method of ensembling to model uncertainty in prediction output:

```
──────────────── Estimate Uncertainty ────────────────
from uncertainty import uncertainty

# Set uncertainty model
uncertainty_model_name = 'ensemble'

# Train uncertainty model
uncertainty_model = uncertainty(uncertainty_model_name,
                                model_parameters, pred_class, task)
uncertainty_model.fit(dataset_training)

# Return uncertainty of the trained predictive model
test_ci_hat = uncertainty_model.predict(dataset_testing)

print('Finish uncertainty estimation')
```

### E.8 INTERPRET PREDICTIONS

Compute feature importance weights (which we name '`test_s_hat`' below). In this example, we use the method of (temporal) INVASE to model instance-wise feature/temporal importance weights:

```
─── Interpret Predictions ───
from interpretation import interpretation

# Set interpretation model
interpretation_model_name = 'tinvase'

# Train interpretation model
interpretor = interpretation(interpretation_model_name,
                             model_parameters, pred_class, task)
interpretor.fit(dataset_training)

# Return instance-wise temporal and static feature importance
test_s_hat = interpretor.predict(dataset_testing)

print('Finish model interpretation')
```

### E.9 VISUALIZE RESULTS

Here we visualize the performance of the trained model (using the `print_performance` method):

```
─── Visualize Performance ───
from evaluation import Metrics
from evaluation import print_performance

# Evaluate predictor model
result = Metrics(metric_sets, metric_parameters).evaluate(
  dataset_testing.label, test_y_hat)
print('Finish predictor model evaluation.')

print('Overall performance')
print_performance(result, metric_sets, metric_parameters)
```

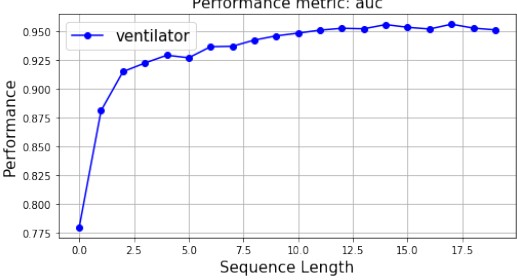

Similar methods can be used to visualize model predictions, uncertainties, and importances (by importing `print_prediction`, `print_uncertainty`, and `print_interpretation` methods). See Jupyter notebook tutorial for complete sample code, including inputs, outputs, and visualizations:

```
─── Other Visualizations ───
from evaluation import print_prediction, print_uncertainty, print_interpretation

# Set the patient index for visualization
index = [1]

print('Each prediction')
print_prediction(test_y_hat[index], metric_parameters)

print('Uncertainty estimations')
print_uncertainty (test_y_hat[index], test_ci_hat[index], metric_parameters)

print('Model interpretation')
print_interpretation (test_s_hat[index], dataset_training.feature_name,
                      metric_parameters, model_parameters)
```

## F  WORKED EXAMPLE: USING THE AUTOML INTERFACE

This section gives a fully worked example of using the Clairvoyance optimization interface (via the treatment effects pathway). Here the basic structure remains the same as in Section E, but in the model training step (here we use CRN for the treatment effects model) we show an example of performing stepwise model selection (SMS) as well. We assume the reader is familiar with the details as in Section E, and do not repeat similar descriptions. Instead, we organize the code as in a standard experiment—using a 'main' function wrapper with top-level arguments to enable ease of inspection.

Import necessary packages, begin main function, and set basic parameters:

```
from __future__ import absolute_import
from __future__ import division
from __future__ import print_function

import argparse
import numpy as np
import warnings; warnings.filterwarnings('ignore')
import sys; sys.path.append('../')

from datasets import CSVLoader
from preprocessing import FilterNegative, OneHotEncoder, Normalizer, ProblemMaker
from imputation import Imputation
from feature_selection import FeatureSelection
from treatments.CRN.CRN_Model import CRN_Model
from prediction import AutoEnsemble
from automl.model import AutoTS
from evaluation import Metrics, BOMetric
from evaluation import print_performance, print_prediction
from utils import PipelineComposer
```

──────── Begin Main Function ────────
```
def main (args):
  '''Args:
    - data loading parameters:
      - data_names: mimic, ward, cf, mimic_antibiotics
    - preprocess parameters:
      - normalization: minmax, standard, None
      - one_hot_encoding: input features that need to be one-hot encoded
      - problem: 'one-shot' or 'online'
        - 'one-shot': one time prediction at the end of the time-series
        - 'online': preditcion at every time stamps of the time-series
      - max_seq_len: maximum sequence length after padding
      - label_name: the column name for the label(s)
      - treatment: the column name for treatments
    - imputation parameters:
      - static_imputation_model: mean, median, mice, missforest, knn, gain, etc.
      - temporal_imputation_model: mean, median, linear, quadratic, cubic, spline, etc.
    - feature selection parameters:
      - feature_selection_model: greedy-addtion, recursive-addition, etc.
      - feature_number: selected feature number
    - predictor_parameters:
      - epochs: number of epochs
      - bo_itr: bayesian optimization iterations
      - static_mode: how to utilize static features (concatenate or None)
      - time_mode: how to utilize time information (concatenate or None)
      - task: classification or regression
    - metric_name: auc, apr, mae, mse
  '''
  # Set basic parameters
  metric_sets = [args.metric_name]
  metric_parameters =  {'problem': args.problem, 'label_name': [args.label_name]}
```

## F.1 LOAD DATASET

```
──────────────────────── Load Dataset ────────────────────────
 # (continued within 'def main')

   # File names
   data_directory = '../datasets/data/' + args.data_name + '/' + args.data_name + '_'

   data_loader_training = \
     CSVLoader(static_file=data_directory + 'static_train_data.csv.gz',
               temporal_file=data_directory + 'temporal_train_data_eav.csv.gz')

   data_loader_testing = \
     CSVLoader(static_file=data_directory + 'static_test_data.csv.gz',
               temporal_file=data_directory + 'temporal_test_data_eav.csv.gz')

   dataset_training = data_loader_training.load()
   dataset_testing = data_loader_testing.load()

   print('Finish data loading.')
```

## F.2 PREPROCESS DATASET

```
──────────────────────── Preprocess Dataset ────────────────────────
 # (continued within 'def main')

   # (0) filter out negative values (Automatically)
   negative_filter = FilterNegative()

   # (1) one-hot encode categorical features
   onehot_encoder = OneHotEncoder(one_hot_encoding_features=[args.one_hot_encoding])

   # (2) Normalize features: 3 options (minmax, standard, none)
   normalizer = Normalizer(args.normalization)

   filter_pipeline = PipelineComposer(negative_filter, onehot_encoder, normalizer)

   dataset_training = filter_pipeline.fit_transform(dataset_training)
   dataset_testing = filter_pipeline.transform(dataset_testing)

   print('Finish preprocessing.')
```

## F.3 DEFINE PROBLEM

```
──────────────────────── Define Problem ────────────────────────
 # (continued within 'def main')

   problem_maker = \
     ProblemMaker(problem=args.problem, label=[args.label_name],
                  max_seq_len=args.max_seq_len, treatment=[args.treatment])

   dataset_training = problem_maker.fit_transform(dataset_training)
   dataset_testing = problem_maker.fit_transform(dataset_testing)

   print('Finish defining problem.')
```

## F.4 IMPUTE DATASET

```
──────────────────────── Impute Dataset ────────────────────────
 # (continued within 'def main')

   static_imputation = Imputation(
     imputation_model_name=args.static_imputation_model, data_type='static')
```

```
   temporal_imputation = Imputation(
     imputation_model_name=args.temporal_imputation_model, data_type='temporal')

   imputation_pipeline = PipelineComposer(static_imputation, temporal_imputation)

   dataset_training = imputation_pipeline.fit_transform(dataset_training)
   dataset_testing = imputation_pipeline.transform(dataset_testing)

   print('Finish imputation.')
```

## F.5 FEATURE SELECTION

```
——————————————————————— Feature Selection ———————————————————————
 # (continued within 'def main')

   static_feature_selection = FeatureSelection(
     feature_selection_model_name=args.static_feature_selection_model,
     feature_type='static',
     feature_number=args.static_feature_selection_number,
     task=args.task,
     metric_name=args.metric_name,
     metric_parameters=metric_parameters)

   temporal_feature_selection = FeatureSelection(
     feature_selection_model_name=args.temporal_feature_selection_model,
     feature_type='temporal',
     feature_number=args.temporal_feature_selection_number,
     task=args.task,
     metric_name=args.metric_name,
     metric_parameters=metric_parameters)

   feature_selection_pipeline = PipelineComposer(static_feature_selection,
                                                 temporal_feature_selection)

   dataset_training = feature_selection_pipeline.fit_transform(dataset_training)
   dataset_testing = feature_selection_pipeline.transform(dataset_testing)

   print('Finish feature selection.')
```

## F.6 OPTIMIZATION AND PREDICTION

Since we want to do stepwise model selection, this step differs from that in Section E. In particular, here we are not just relying on a single pathway model (CRN); we are calling the 'AutoTS' module to perform Bayesian optimization—which implements SMS by DKL, exactly as described in [71]:

```
——————————————————— Optimization and Prediction ———————————————————
 # (continued within 'def main')

   # CRN model
   model_parameters = {'projection_horizon': 5,
                       'encoder_max_alpha':1,
                       'decoder_max_alpha': 1,
                       'static_mode': 'concatenate',
                       'time_mode': 'concatenate'}

   crn_model = CRN_Model(task='classification')
   crn_model.set_params(**model_parameters)

   model_class= crn_model

   # train_validate split
   dataset_training.train_val_test_split(prob_val=0.2, prob_test=0.2)
```

```
    # Bayesian Optimization Start
    metric = BOMetric(metric='auc', fold=0, split='test')

    # Run BO for selected model class
    BO_model = AutoTS(dataset_training, model_class, metric)
    models, bo_score = BO_model.training_loop(num_iter=20)
    auto_ens_model = AutoEnsemble(models, bo_score)

    # Prediction of treatment effects
    test_y_hat = auto_ens_model.predict(dataset_testing, test_split='test')

    print('Finish AutoML model training and testing.')
```

## F.7  MODEL EVALUATION

Output performance evaluation for the final trained model, using metric parameters as defined above:

```
──────── Model Evaluation ────────
 # (continued within 'def main')

    result = Metrics(metric_sets, metric_parameters).evaluate(test_y, test_y_hat)
    print('Finish ITE model evaluation.')

    print('Overall performance')
    print_performance(result, metric_sets, metric_parameters)
```

## F.8  TOP-LEVEL PARSER

```
──────── Define and Parse Arguments ────────
if __name__ == '__main__':
  parser = argparse.ArgumentParser()
  parser.add_argument(
    '--data_name',
    choices=['mimic', 'ward', 'cf', 'mimic_antibiotics'],
    default='mimic_antibiotics',
    type=str)
  parser.add_argument(
    '--normalization',
    choices=['minmax', 'standard', None],
    default='minmax',
    type=str)
  parser.add_argument(
    '--one_hot_encoding',
    default='admission_type',
    type=str)
  parser.add_argument(
    '--problem',
    choices=['online', 'one-shot'],
    default='online',
    type=str)
  parser.add_argument(
    '--max_seq_len',
    help='maximum sequence length',
    default=20,
    type=int)
  parser.add_argument(
    '--label_name',
    default='ventilator',
    type=str)
  parser.add_argument(
    '--treatment',
    default='antibiotics',
    type=str)
```

```
parser.add_argument(
  '--static_imputation_model',
  choices=['mean', 'median', 'mice', 'missforest', 'knn', 'gain'],
  default='median',
  type=str)
parser.add_argument(
  '--temporal_imputation_model',
  choices=['mean', 'median', 'linear', 'quadratic', 'cubic', 'spline',
           'mrnn', 'tgain'],
  default='median',
  type=str)
parser.add_argument(
  '--static_feature_selection_model',
  choices=['greedy-addition', 'greedy-deletion', 'recursive-addition',
           'recursive-deletion', None],
  default=None,
  type=str)
parser.add_argument(
  '--static_feature_selection_number',
  default=10,
  type=int)
parser.add_argument(
  '--temporal_feature_selection_model',
  choices=['greedy-addition', 'greedy-deletion', 'recursive-addition',
           'recursive-deletion', None],
  default=None,
  type=str)
parser.add_argument(
  '--temporal_feature_selection_number',
  default=10,
  type=int)
parser.add_argument(
    '--epochs',
    default=20,
    type=int)
parser.add_argument(
    '--bo_itr',
    default=20,
    type=int)
parser.add_argument(
    '--static_mode',
    choices=['concatenate',None],
    default='concatenate',
    type=str)
parser.add_argument(
    '--time_mode',
    choices=['concatenate',None],
    default='concatenate',
    type=str)
parser.add_argument(
    '--task',
    choices=['classification','regression'],
    default='classification',
    type=str)
parser.add_argument(
    '--metric_name',
    choices=['auc','apr','mse','mae'],
    default='auc',
    type=str)

# Call main function
args = parser.parse_args()
main(args)
```

## G  EXTENSIBILITY: EXAMPLE WRAPPER CLASS

Since novel methods are proposed in the ML community every day, the pipeline components should be easily extensible to incorporate new algorithms. To integrate a new component method (e.g. from another researcher's code, or from an external package) into the framework, all that is required is a simple wrapper class that implements the 'fit', 'predict', and 'get-hyperparameter-space' methods. Here we show an example of how a classical time-series prediction model (ARIMA) can be integrated.

```
──── ARIMA Wrapper Class ────
# Necessary packages
import os
import pmdarima as pm
from datetime import datetime
from base import BaseEstimator, PredictorMixin
import numpy as np

class ARIMA(BaseEstimator, PredictorMixin):
  """Attributes:
    - task: classification or regression
    - p: MA degree
    - d: Differencing degree
    - q: AR degree
    - time_mode: 'concatenate' or None
    - model_id: the name of model
    - model_path: model path for saving
    - verbose: print intermediate process
  """
  def __init__(self,
               task=None,
               p=None,
               d=None,
               q=None,
               time_mode=None,
               model_id='auto_arima',
               model_path='tmp',
               verbose=False):

    super().__init__(task)

    self.task = task
    self.p = p
    self.d = d
    self.q = q
    self.time_mode = time_mode
    self.model_path = model_path
    self.model_id = model_id
    self.verbose = verbose

    # Predictor model & optimizer define
    self.predictor_model = None

    if self.task == 'classification':
      raise ValueError('Arima model cannot be used for classification')

    # Set path for model saving
    if not os.path.exists(model_path):
      os.makedirs(model_path)
    self.save_file_name = '{}/{}'.format(model_path, model_id) + \
      datetime.now().strftime('%H%M%S') + '.hdf5'

  def new(self, model_id):
    """Create a new model with the same parameter as the existing one.
    Args:
      - model_id: an unique identifier for the new model
```

```
    Returns:
      - a new ARIMA
    """
    return ARIMA(self.task,
                 self.p,
                 self.d,
                 self.q,
                 self.time_mode,
                 model_id,
                 self.model_path,
                 self.verbose)
```

──────────────────────────────── fit ────────────────────────────────
```
def fit(self, dataset, fold=0, train_split='train', valid_split='val'):
    """ Arima model fitting does not require an independent training set.
    """
    pass
```

──────────────────────────────── predict ────────────────────────────────
```
def predict(self, dataset, fold=0, test_split='test'):
    """"Return the predictions based on the trained model.
    Args:
      - dataset: temporal, static, label, time, treatment information
      - fold: Cross validation fold
      - test_split: testing set splitting parameter
    Returns:
      - test_y_hat: predictions on testing set
    """
    test_x, test_y = self._data_preprocess(dataset, fold, test_split)
    shape0 = test_y.shape[0]
    shape1 = test_y.shape[1]
    shape2 = test_y.shape[2]

    print(test_y.shape)

    assert shape2 == 1

    # y: N_sample, max_seq_len, dim
    fited_list = []

    for i in range(shape0):
      y0 = test_y[i, :, 0]
      model = pm.arima.ARIMA(order=(self.p, self.d, self.q), suppress_warnings=True)
      try:
        model.fit(y0)
        y_hat = model.predict_in_sample(dynamic=True)
      except Exception:
        y_hat = np.zeros_like(y0)
      fited_list.append(y_hat)

    y_hat = np.stack(fited_list, axis=0)[:, :, None]
    return y_hat

  @staticmethod
```

──────────────────────────────── get_hyperparameter_space ────────────────────────────────
```
def get_hyperparameter_space():
  hyp_ = [{'name': 'p', 'type': 'discrete', 'domain': list(range(1, 6)), 'dimensionality': 1},
          {'name': 'd', 'type': 'discrete', 'domain': list(range(1, 6)), 'dimensionality': 1},
          {'name': 'q', 'type': 'discrete', 'domain': list(range(1, 6)), 'dimensionality': 1}]
  return hyp_
```

## H   SOME FREQUENTLY ASKED QUESTIONS

**Q1**. Does Clairvoyance include every time-series model under the sun?

**A1**. That is not our purpose in providing the pipeline abstraction (see Section 2: "As a Software Toolkit"), not to mention generally impossible. We do include standard classes of models (e.g. popular deep learning models for prediction), and an important contribution is in unifying all three key tasks involved in a patient's healthcare lifecycle under a single roof, including the treatment effects pathway and active sensing pathway (both for which we provide state-of-the-art time-series models) in addition to the predictions pathway (see Section 2: "The Patient Journey", Figures 1–2, as well as Table 1). Moreover, as noted throughout, modules are easily extensible: For instance, if more traditional time-series baselines from classical literature are desired for comparison purposes, existing algorithms from [73–79] can be integrated into Clairvoyance by using wrappers, with little hassle.

**Q2**. Isn't preprocessing, imputation, selection, etc. already always performed?

**A2**. Yes, and we are not claiming that there is anything wrong with individual studies per se. However (per Section 2: "As an Empirical Standard", and Appendix A: Tables 6–7), while current research practices typically seek to isolate individual gains, the degree of clarity and/or overlap in pipeline configurations across studies is lacking. This dearth of empirical standardization may not optimally promote practical assessment/reproducibility, and may obscure/entangle true progress. By providing a software toolkit and empirical standard, constructing an end-to-end solution to each problem is easy, systematic, and self-documenting (see Figures 2–3), and evaluating collections of models by varying a single component ensures that comparisons are standardized, explicit, and reproducible.

**Q3**. How about other issues like regulations, privacy, and federated learning?

**A3**. Per the discussion in Section 3, Clairvoyance is not a solution for preference-/application-specific considerations such as cohort construction, data cleaning and heterogeneity, patient privacy, algorithmic fairness, federated learning, or compliance with government regulations. While such issues are real/important concerns (with plenty of research), they are firmly beyond the scope of our software; it is designed to operate in service to clinical decision support—not at all to replace humans in the loop.

**Q4**. What are these interdependencies among components and time steps?

**A4**. Componentwise interdependencies occur for any number of reasons. We have discussed several examples (see Section 2: "As an Empirical Standard"), but it is not our mission to convince the reader from scratch: For that, there exists a plethora of existing autoML/medical literature (see e.g. Section 3). However, the pipeline abstraction serves as a succinct and standardized interface to anyone's favorite autoML algorithm (see Section 2: "As an Optimization Interface"). Moreover, here we do specifically highlight the temporal dimension of model selection opened up by the time-series nature of the pipeline (see Figure 5). In particular, each example in Section 4 specifically illustrates the gains in performance that already occur—ceteris paribus—using a simple approach to SASH as in Figure 4(a).

**Q5**. Where is all the background and theory on each module?

**A5**. The scope of the software toolkit is purposefully broad, but it is not our intention to provide a technical introduction to each of the topics involved (which would—in any case—be impossible in the scope of a paper). While Clairvoyance lowers the barrier to entry in terms of engineering/evaluation, it is not intended to be used as a black-box solution. For instance, we expect that a user desiring to conduct treatment effects estimation using the CRN component to be familiar with its basic theory and limitations. That said, in addition to the various references provided throughout the description of each aspect of Clairvoyance, the following may serve as more concise background information on original problem formulations and solutions: For treatment effects estimation over time we refer to [31]; for active sensing we refer to [34]; for time-series data imputation we refer to [15]; for interpretation by individualized variable selection we refer to [44]; for autoML in general we refer to [63]; for the pipeline configuration and selection (PSC) problem we refer to Section 3.1 in [67]; and for the stepwise model selection (SMS) problem we refer to Sections 2–3 in [71]; moreover, Figure 5 shows how new problems (e.g. SASH) directly result from combining their optimization domains.

**Q6**. How do you know what clinicians want?

**A6**. With clinicians as developers/authors, it is our central goal to understand realistic usage scenarios.

# I  GLOSSARY OF ACRONYMS

**ASAC**: Active sensing by actor critic, first defined in [36].

**APR**: Area under the precision-recall curve.

**AUC**: Area under the receiver-operating characteristic curve.

**CASH**: Combined algorithm selection and hyperparameter optimization, first defined in [63].

**CRN**: Counterfactual recurrent network, first defined in [31].

**DKL**: Deep kernel learning, first defined in [72].

**FLASH**: Fast linear search, first defined in [68].

**GAIN**: Generative adversarial imputation network, first defined in [97].

**GANITE**: Generative adversarial network for individualized treatment effects, first defined in [98].

**GRU**: Gated Recurrent Units, a type of recurrent neural network.

**INVASE**: Instance-wise variable selection, first defined in [44].

**LSTM**: Long-short term memory, a type of recurrent neural network.

**MAE**: Mean absolute error.

**MICE**: Multiple imputation by chained equations.

**MissForest**: Missing value imputation using random forest.

**MRNN**: Multi-directional recurrent neural networks, first defined in [15].

**PSC**: Pipeline selection and configuration, first defined in [67].

**RMSE**: Root mean squared error.

**RMSN**: Recurrent marginal structural network, first defined in [30].

**RPS**: Relaxed parameter sharing, first defined in [53].

**SASH**: Stepwise algorithm selection and hyperparameter optimization, first defined in Section 2.

**SKL**: Structured kernel learning, first defined in [69].

**SMS**: Stepwise model selection, first defined in [71].

**TCN**: Temporal convolutional network.

*Note*: A prefix of "T" to certain techniques simply indicates its temporal counterpart (e.g. "T-GAIN" refers to the method of GAIN using a recurrent neural network for handling the temporal dimension).

