# OpenReview forum: "Clairvoyance: A Pipeline Toolkit for Medical Time Series"
_ICLR.cc/2021/Conference — ICLR 2021 Poster_

### Official Review · AnonReviewer4 · 2020-10-28
**Timely project for healthcare application - some concern about the study design**

**Rating:** 8
**Confidence:** 5

**Review:**

Very interesting and timely project.

Major concern:
For a well-planned model development with large enough dataset, it is recommended to separate the case and controls from the very beginning and apply the pre-processing and imputation on the training dataset only. Feature selection should also be based on the training data alone, which is not the case in this pipeline from my understanding. The way, the pipeline is described, the processing, including imputation and feature selection are performed on the full dataset, which is then passed to the modeling phase. During the modeling phase, where model training and validation and testing will be performed.

This can be cause of over-fitting since the testing dataset was to some level "seen" before the model testing. This has to be stated in the limitation of the study design.

Having a team science approach with clinician scientists as part of the team is integral part of the study, which seems that this paper is all about.



Minor:
Some grammatical /stylistic error. ex: “While issues such as data cleaning, algorithmic fairness, and privacy and heterogeneity have import, they are beyond the scope of our software.”  revise the sentence “have import”.


Finally, I am not a software engineer and will leave that level of evaluation to my colleagues.

---

> ### Author Response · Authors · 2020-11-19
> **Response to Reviewer #4**
>
> ---
>
> Thank you for your thoughtful comments and suggestions.
>
> ---
>
> **(1) Separation of Training and Testing Sets**
>
> We completely agree that preprocessing, imputation, and feature selection should be based on the training data alone, such that there is no “data leakage” going on. Actually, the pipeline is designed such that we *do* always adhere to follow this (very important) principle. Kindly allow us to clarify:
>
> In each of these “processing” modules, there are two distinct methods provided (and similarly, any new user-created module would need to expose these two methods):
>
> - "fit_transform". This is applied to the training data. When this is called on the training data, the “processing” is *based on* the training data, and then *applied to* the training data.
>
> - "transform". This is applied to the testing data. When this is called on the testing data, the “processing” is still just the one based *solely* on the training data; we are just *applying* it to the testing data.
>
> In this manner, no overfitting due to “data leakage” can occur. Importantly, note that this “fit_transform / transform” paradigm that we adopt is exactly as is standard in practice (see e.g. sklearn’s interface).
>
> Furthermore, in order to better illustrate this important principle in a step-by-step manner (and prevent inadvertent misunderstanding), we have now included in the revised manuscript a (new) Appendix E, which provides a fully-worked example of using the pipeline to train and predict with a model (for this, we use the predictions pathway). This takes the form of a detailed, step-by-step walk-through of the entire pipeline and its components, with fully-executable code, comments, and accompanying descriptions where appropriate.
>
> ---
>
> **(2) Grammatical Error**
>
> Thank you for pointing out the grammatical error---we agree, and have revised the sentence as follows:
>
> “[...] and privacy and heterogeneity are important, [...]”
>
> ---
>
> With our clarifications and revisions, we hope that we have addressed your concerns. Thank you for your kind consideration.

---

### Official Review · AnonReviewer1 · 2020-10-28
**This paper provides a new software package to standardize many common tasks associated with clinical time series data, along with some illustrative examples**

**Rating:** 4
**Confidence:** 4

**Review:**

##########################################################################

Summary:

The authors present a new package aimed at improving the design and validation of pipelines using medical time series data. The pipeline covers many aspects of time series pipelines including pre-processing, prediction, treatment effect estimation, calibration, etc. The package, as depicted in the paper, appears to be very comprehensive and well motivated.

##########################################################################

Rationale for score:

The primary reason for my recommendation of reject is this paper is ill-suited for a venue such as ICLR. The paper is a descriptive paper for a new software package and it is aimed at the healthcare/informatics sub-community. While I appreciate the considerable effort the authors have clearly put into this software package, I think the paper would have a better chance of reaching it's intended audience in a more focused venue such as the Journal of the American Medical Informatics Association (JAMIA), the Machine Learning for Healthcare Conference (MLHC), or the Journal of Statistical Software (JSS). I also believe that the ICLR format does a disservice to the authors, as 8 pages is not enough room to fully elaborate on their work and the current version is very compressed which makes for difficult reading. Given the space constraints, the paper feels more like an advertisement for the package vs. an elaboration and explanation document. I think there is an impressive amount of work on display here, but I think that ultimately a paper such as this would be better served in a different venue.

##########################################################################

Pros:

1. The pipeline, as it is proposed in the paper is quite impressive. Moreover, the authors do an excellent job at motivating the need for such a pipeline not only as a tool but as a means of standardization and benchmarking, something that is sorely needed in many healthcare applications of machine learning.

2. The figures and table 1 do a good job at summarizing the proposed approach and existing alternatives.

##########################################################################

Cons:

1. There are no empirical evaluations against alternative methods in this paper. At minimum there should be some kind of head to head evaluations against the existing packages.

2. I found the vignettes too condensed to be helpful and it's unclear how the proposed pipeline was used to produce the results in the tables. The authors do a good job at setting up the clinical problem, but (likely due to space constraints) it is unclear how the problem reduces to a series of pipeline steps. Having one fully worked example with code (even in the appendix) would greatly help to understand how the proposed pipeline works.

3. Several basic details are missing from the paper. For example, is this a python package? From the code snippets, I assume it is but this is never stated in the paper. If it is a python package, what versions of python is it compatible with, what are the dependencies, is GPU acceleration supported, etc? I was able to find some of this by digging through the included code but these details should be included in the paper.

4. Many acronyms are used but never defined in the text, e.g. CRN and R-MSN from the examples. If, as the authors claim, they would like their package to be used by clinicians and ML practitioners alike, the should define these acronyms in the text to aid the reader.

5. Are all of these modules complete or are some still in the alpha phase of development?

---

> ### Author Response · Authors · 2020-11-19
> **Response to Reviewer #1 [Part 5/5]**
>
> ---
>
> **References** (numbering as continued from revised manuscript)
>
> [99] ICLR 2020: Novak et al., "Neural Tangents: Fast and Easy Infinite Neural Networks in Python".
>
> [100] ICLR 2020: Osband et al., "BSuite: Behavior Suite for Reinforcement Learning".
>
> [101] ICLR 2019: Schneider et al., "DeepOBS: A Deep Learning Optimizer Benchmark Suite".
>
> [102] ICML 2020: Goyal et al., "PackIt: A Virtual Environment for Geometric Planning".
>
> [103] ICML 2020: Hu et al., "EXTREME: A Massively Multilingual Multi-task Benchmark for Evaluating Cross-lingual Generalization".
>
> [104] ICML 2019: Bansal et al., "HOList: An Environment for Machine Learning of Higher-Order Theorem Proving".
>
> [105] NeurIPS 2020: Schoenholz et al., "JAX, M.D.: A Framework for Differentiable Physics".
>
> [106] NeurIPS 2019: Tran et al., "Bayesian Layers: A Module for Neural Network Uncertainty".
>
> [107] NeurIPS 2019: Park et al., "Park: An Open Platform for Learning-Augmented Computer Systems".

---

> ### Author Response · Authors · 2020-11-19
> **Response to Reviewer #1 [Part 4/5]**
>
> ---
>
> (C.5) Completeness of development:
>
> As far as the overall *structure* of the pipeline as proposed and presented in the paper, development is fairly complete (save internal refactoring or slight changes in naming conventions, etc).
>
> As for the *scope* of the project (i.e. extending components to incorporate newer/more powerful individual techniques, or even adding additional components based on community feedback), we hope to remain in active development and community engagement on an ongoing basis into the future.
>
> In this latter sense, as an open project, development is ideally never “complete”. In fact, we do have a fully-developed agenda for community engagement and feedback in execution. While we must necessarily be brief on the following details (due to the requirement for anonymity during the review process), this should give an adequate idea of outreach:
>
> - Our lab introduced this framework/initiative in May of this year in an official announcement, accompanying technical description, and publicly available source code.
>
> - In a keynote address at a workshop in a prominent ML conference in July, this framework was introduced, and participants were encouraged to download, try out, and give feedback on our software.
>
> - In August, a piece of long-form content was published by the team, highlighting the framework and calling for readers to visit the software repository.
>
> Moreover, from September to the present (and continuously going forward), we introduced the framework through a series of our lab’s hour-long online “engagement sessions” for machine learning and healthcare researchers and students. Specifically:
>
> - In September, a general introduction to the pipeline abstraction was given, while participants were encouraged to visit our software repository and try the framework in their own projects.
>
> - In October, our engagement session featured a 10-minute section specifically dedicated to explaining the mission of the initiative, as well as the goals of the framework.
>
> - Most recently, in November our engagement session focused extensively on the framework--including 16 minutes of tutorials on how to use the software, and how to add new algorithms to the framework, as well as fielding numerous questions on the pipeline during the subsequent Q&A session.
>
> In time, these initiatives will lead us towards more mature community-driven development, whereupon we will be equipped to measure and report user interviews, feedback, and adoption patterns.
>
> ---
>
> With our clarifications and revisions, we hope that we have addressed your concerns. Thank you for your kind consideration.

---

> ### Author Response · Authors · 2020-11-19
> **Response to Reviewer #1 [Part 3/5]**
>
> ---
>
> **(C) Miscellaneous Questions**
>
> (C.1) Evaluations against existing packages:
>
> Kindly allow us to reiterate that the proposed framework is the first of its kind, and there are no other frameworks that provide an end-to-end, unified structure to the *entire* time-series workflow (see Section 4: “Related Work”, and Table 1: “Clairvoyance and Comparable Software”). While there is existing software providing *implementations* of individual models and optimization algorithms [73--79], there is no comparable *framework-level* software that makes sense to “juxtapose” with (especially in the medical setting)---in fact, this is precisely what motivated developing/disseminating Clairvoyance in the first place.
>
> Moreover, it goes without saying that implementation-based packages such as [73--79] exist in *complement* to the Clairvoyance pipeline (rather than in competition)---especially as the worked example in Appendix G now shows. Likewise, this is also true of existing research for model selection: For instance, as the worked example in Appendix F now shows, it is precisely the pipeline abstraction and proposed optimization interface that allows us to “plug in” the existing technique of deep kernel learning (DKL) in tackling the stepwise model selection (SMS) problem.
>
> (C.2) Fully worked example with code:
>
> We agree that worked examples do greatly benefit clarity of exposition, and we have indeed now included a number of such coded examples in the revised Appendices. Please refer our much more detailed response (B.1) above: We now include a complete worked example of using the pipeline (Appendix E), an example of using the stepwise optimization interface (Appendix F), and example of how to extend component modules to include a new technique (Appendix G). In addition, please also refer to our response (B.2) above: We now tabulate more background and references to original studies that provide more concrete information about the medical and theoretical relevance (i.e. as ML modeling problems) of our illustrative examples.
>
> (C.3) Basic software details:
>
> Indeed, Clairvoyance is a *python* package. We completely agree that this should be stated up front. We have now updated the cover page to include “Python” in front of the (placeholder) link to the repository.
>
> We also agree that versioning and dependencies should be more accessible. In the camera-ready version (where anonymity is no longer an issue), we shall include the link to the software repository and license on the very first page. That way, the official “readme” and “requirements.txt” (containing information about package versions and dependencies) will be one click away. (Since Clairvoyance will remain under active development and community engagement, these details may need to be updated rapidly, so a direct link to the software repository is easier to maintain than a static Appendix entry. See also point (C.5) below).
>
> Finally, allow us to clarify that as a *high-level* API framework, support for GPU acceleration simply depends on the underlying implementation details of the component modules being used. For the deep learning models that are already built into the initial version of Clairvoyance, this is certainly supported. (However, if an external model is integrated via a wrapper class, for instance, then this depends entirely on how that external model is implemented).
>
> (C.4) Definitions of acronyms:
>
> Thank you for pointing out that some acronyms are not properly defined. We agree that---for better accessibility---these should be identifiable within the paper itself.
>
> For optimal referencing, in a (new) Appendix I in the revised manuscript, we now provide a list of all potentially lesser-known acronyms. Where appropriate, we also provide the original citation to the paper in which the acronym is first defined. For instance, for the two examples mentioned:
>
> CRN: Counterfactual recurrent network, first defined in [31].
>
> R-MSN: Recurrent marginal structural network, first defined in [30].
>
> etc.

---

> ### Author Response · Authors · 2020-11-19
> **Response to Reviewer #1 [Part 2/5]**
>
> ---
>
> **(B) Elaboration and Explanation**
>
> Thank you for pointing out the importance of balancing between “advertising” and more thorough “elaboration and explanation” in the paper. We agree that the paper's exposition would benefit from more *concrete* details. To this end, in the revised manuscript we have made a number of important additions to provide a complete picture of our mission, contribution, and actual usage of the platform.
>
> (B.1) Fully-Worked Code Examples:
>
> Further speaking to our ideals of reproducibility and transparency, we now provide *fully worked examples* of Clairvoyance in actual usage, which directly demonstrates how an experiment is exactly run, as well as further highlighting the simplicity and standardization of coding with the pipeline abstraction.
>
> (a) In the (new) Appendix E in the revised manuscript, we now provide a worked example of using the *Clairvoyance pipeline* to train and use a model (for this, we use the predictions pathway). This takes the form of a detailed, step-by-step walk-through of the entire pipeline and its components, with fully-executable code, comments, and accompanying descriptions where appropriate.
>
> (b) In the (new) Appendix F in the revised manuscript, we now provide a worked example of using the *optimization interface* to perform stepwise model selection (for this, we use the treatment effects pathway for variety). This also takes the form of executable, commented code, organized as in a standard experiment---using a “main” function wrapper with top-level arguments.
>
> (c) In the (new) Appendix G in the revised manuscript, we now provide an example of how a generic *wrapper class* should be written for the purpose of integrating an external model/algorithm that is not already implemented in the current version of Clairvoyance. Specifically, we show an example of how a classical time-series prediction model (ARIMA) can be easily integrated.
>
> In the main text of the revised manuscript, pointers to all of these worked examples are now included in a dedicated “focus box” at the end of Section 4 with its own subtitle: “Worked Examples”. Furthermore, this also includes references to the interactive tutorials (using Jupyter notebooks) and top-level API code with examples of pathways and optimizations, that are part of the software repository.
>
> (B.2) Background on Illustrative Examples:
>
> In the revised manuscript, we have now included a (new) Appendix B: “Background References for Experiments”, which tabulates exemplary references to original papers for the following details on each modeling question (e.g. “clinical deterioration of ward patients”), as well as---where appropriate---pointers to the relevant sections in those papers that we follow.
>
> - original works describing and justifying the modeling task,
>
> - original description/documentation for the dataset used, including
>
> - details on data collection, coverage, ethics, etc.,
>
> - the procedure for initial data cleaning/selection that we followed, as well as
>
> - theoretical background on the Clairvoyance pathway involved in the modeling question.
>
> Putting it all together, we believe that the combination of the following elements now adequately demonstrates the motivation, contribution, and ease of adoption of the Clairvoyance pipeline: Section 4 (high-level motivations and medically relevant use cases), Appendix B (background and original references), Appendix D (low-level experiment details), and Appendices E, F, G (fully-worked/executable examples).

---

> ### Author Response · Authors · 2020-11-19
> **Response to Reviewer #1 [Part 1/5]**
>
> ---
>
> Thank you for your thoughtful comments and suggestions. We give answers to each in turn, as well as pointing out corresponding updates to the revised manuscript (additions are made in *blue*). Sections (A) and (B) below are aimed at the “rationale for score”, and Section (C) is aimed at the miscellaneous questions.
>
> ---
>
> **(A) Software Platform and ICLR Venue**
>
> Thank you for bringing up the question of venue. We would like to take care in pointing out the following, in order to gently clarify the appropriateness of the submission for ICLR and our intended audience:
>
> (a) Software platforms are indeed a core part of ML conferences (main conference), although usually not as conspicuous or numerous as theory-/methods-based papers.
>
> - ICLR: The “Call for Papers” at ICLR directly seeks submissions for “Software Platforms” and “Implementation Issues” (this is found under the “List of Relevant Topics”, and has been part of this list for every year since the very first ICLR conference in 2013).
>
> - NeurIPS and ICML: Likewise for other similar conferences, where the “Call for Papers” directly seeks submissions for “Implementations and Software” and “Software Toolkits” (Item #3) at NeurIPS and “Applications” (Item #7) for ICML (also since the past).
>
> (b) While not as prominent, there have been many such contributions. For a random selection of similar initiatives published at ML conferences (main conference track) in the past two years, we refer to [99--101] for ICLR, [102--104] for ICML, and [105--107] for NeurIPS. Note that although the specific ML subfields are different from ours, these all similarly propose high-level API software packages and/or standardized environments to enable rapid prototyping and ease of development, as well as to encourage encourage systematic and reproducible validation. Crucially, while some of these are highly specialized (e.g. molecular physics in [105], cross-lingual transfer in [103], etc.), their focus---like ours---is on what the *ML-side* of the cross-disciplinary community can do to further the initiative.
>
> (c) Finally, at risk of repetition, we reiterate that Clairvoyance is *not* designed as a preset collection of algorithms to be repeatedly applied by end-users. Indeed, purely implementation-focused packages such as [73--79] may not find their place in this venue, versus a more specialized journal. But quite to the contrary, Clairvoyance is a *framework* developed as part of our mission to tackle the paramount issues of reproducibility, standardization, and transparency in (cross-disciplinary, collaborative) ML research. This paper is therefore part software description, and part “call-to-arms” for the *ML/DL research community* itself to strive for these ideals in the context of medical time series, much as [103] and [105] do for theirs, for instance. In order to reach the intended audience, we therefore firmly believe that a venue such as ICLR is much more appropriate than a medical journal.

---

> ### Author Response · Authors · 2020-11-25
> **Dear Reviewer #1**
>
> ---
>
> We are sincerely grateful for your time and energy in the review process.
>
> In light of our responses (Nov 19) and revisions (Nov 19), we would appreciate if the reviewer kindly let us know of any leftover concerns in the very limited time remaining. We would be happy to do our utmost to address them.
>
> Thank you!
>
> Paper3220 Authors

---

### Official Review · AnonReviewer3 · 2020-10-31
**Review of the submission called Clairvoyance: A Pipeline Toolkit for Medical Time Series**

**Rating:** 6
**Confidence:** 4

**Review:**

The manuscript introduces and illustrates an end-to-end software pipeline, called Clairvoyance, for medical machine learning on time-series data. The authors must be congratulated for having designed and developed this wonderful resource to accelerate the adoption of these computational techniques in clinical practice as a way to support people’s judgement and decision-making. The manuscript excels in describing and relating its contributions with related work. It has also included a convincing set of experimentation on datasets from three medical environments that are supplementary to each other. My only concerns with this paper are (i) describing and justifying these experiments, their materials (also research ethics), and their processing, evaluation, and statistical significance testing methods to an extent that allows the reader to comprehend these original studies that are now a part of the pipeline release paper (perhaps an appendix or references to separate original studies would do), (ii) release details of the pipeline seem to be missing (e.g., where to get the code, what is the licence of the release, and how are the authors facilitating people adopting the toolkit), and (iii) I am uncertain if ICLR is the right venue for the manuscript to obtain envisioned impacts of the conclusion section of the paper — I would have seen this contribution published in a medical journal instead, and the lacking details related to the item (ii) before make it even harder to assess this paper. However, the submission is excellent, and the program committee should discuss this case further.

---

> ### Author Response · Authors · 2020-11-19
> **Response to Reviewer #3 [Part 4/4]**
>
> ---
>
> **References** (numbering as continued from revised manuscript)
>
> [99] ICLR 2020: Novak et al., "Neural Tangents: Fast and Easy Infinite Neural Networks in Python".
>
> [100] ICLR 2020: Osband et al., "BSuite: Behavior Suite for Reinforcement Learning".
>
> [101] ICLR 2019: Schneider et al., "DeepOBS: A Deep Learning Optimizer Benchmark Suite".
>
> [102] ICML 2020: Goyal et al., "PackIt: A Virtual Environment for Geometric Planning".
>
> [103] ICML 2020: Hu et al., "EXTREME: A Massively Multilingual Multi-task Benchmark for Evaluating Cross-lingual Generalization".
>
> [104] ICML 2019: Bansal et al., "HOList: An Environment for Machine Learning of Higher-Order Theorem Proving".
>
> [105] NeurIPS 2020: Schoenholz et al., "JAX, M.D.: A Framework for Differentiable Physics".
>
> [106] NeurIPS 2019: Tran et al., "Bayesian Layers: A Module for Neural Network Uncertainty".
>
> [107] NeurIPS 2019: Park et al., "Park: An Open Platform for Learning-Augmented Computer Systems".

---

> ### Author Response · Authors · 2020-11-19
> **Response to Reviewer #3 [Part 3/4]**
>
> ---
>
> **(3) Software Platform and ICLR Venue**
>
> Thank you for bringing up the question of venue. We would like to take care in pointing out the following, in order to gently clarify the appropriateness of the submission for ICLR and our intended audience:
>
> (a) Software platforms are indeed a core part of ML conferences (main conference), although usually not as conspicuous or numerous as theory-/methods-based papers.
>
> - ICLR: The “Call for Papers” at ICLR directly seeks submissions for “Software Platforms” and “Implementation Issues” (this is found under the “List of Relevant Topics”, and has been part of this list for every year since the very first ICLR conference in 2013).
>
> - NeurIPS and ICML: Likewise for other similar conferences, where the “Call for Papers” directly seeks submissions for “Implementations and Software” and “Software Toolkits” (Item #3) at NeurIPS and “Applications” (Item #7) for ICML (also since the past).
>
> (b) While not as prominent, there have been many such contributions. For a random selection of similar initiatives published at ML conferences (main conference track) in the past two years, we refer to [99--101] for ICLR, [102--104] for ICML, and [105--107] for NeurIPS. Note that although the specific ML subfields are different from ours, these all similarly propose high-level API software packages and/or standardized environments to enable rapid prototyping and ease of development, as well as to encourage encourage systematic and reproducible validation. Crucially, while some of these are highly specialized (e.g. molecular physics in [105], cross-lingual transfer in [103], etc.), their focus---like ours---is on what the *ML-side* of the cross-disciplinary community can do to further the initiative.
>
> (c) Finally, at risk of repetition, we reiterate that Clairvoyance is *not* designed as a preset collection of algorithms to be repeatedly applied by end-users. Indeed, purely implementation-focused packages such as [73--79] may not find their place in this venue, versus a more specialized journal. But quite to the contrary, Clairvoyance is a *framework* developed as part of our mission to tackle the paramount issues of reproducibility, standardization, and transparency in (cross-disciplinary, collaborative) ML research. This paper is therefore part software description, and part “call-to-arms” for the *ML/DL research community* itself to strive for these ideals in the context of medical time series, much as [103] and [105] do for theirs, for instance. In order to reach the intended audience, we therefore firmly believe that a venue such as ICLR is much more appropriate than a medical journal.
>
> ---
>
> With our clarifications and revisions, we hope that we have addressed your concerns. Thank you for your kind consideration.

---

> ### Author Response · Authors · 2020-11-19
> **Response to Reviewer #3 [Part 2/4]**
>
> ---
>
> **(2) Release Details and Community Engagement**
>
> (2.1) Community Engagement:
>
> Thank you for pointing out the importance of release details and encouraging user adoption. We completely agree that this an important step in the development cycle of any software framework.
>
> To be clear, we do have a fully-developed agenda for *community engagement* in execution. While we are necessarily brief on the following details (due to the requirement for anonymity during the review process), this should give an adequate idea of outreach:
>
> - Our lab introduced this framework/initiative in May of this year in an official announcement, accompanying technical description, and publicly available source code.
>
> - In a keynote address at a workshop in a prominent ML conference in July, this framework was introduced, and participants were encouraged to download, try out, and give feedback on our software.
>
> - In August, a piece of long-form content was published by the team, highlighting the framework and calling for readers to visit the software repository.
>
> Moreover, from September to the present (and continuously going forward), we introduced the framework through a series of our lab’s hour-long online “engagement sessions” for machine learning and healthcare researchers and students. Specifically:
>
> - In September, a general introduction to the pipeline abstraction was given, while participants were encouraged to visit our software repository and try the framework in their own projects.
>
> - In October, our engagement session featured a 10-minute section specifically dedicated to explaining the mission of the initiative, as well as the goals of the framework.
>
> - Most recently, in November our engagement session focused extensively on the framework--including 16 minutes of tutorials on how to use the software, and how to add new algorithms to the framework, as well as fielding numerous questions on the pipeline during the subsequent Q&A session.
>
> In time, these initiatives will lead us towards more mature community engagement, whereupon we will be equipped to measure and report user interviews, feedback, and adoption patterns. Finally, note that in the camera-ready version, the link to the software repository and license will feature on the first page of the paper. (That said, note that a development version of the code is available in the supplementary material for perusal).
>
> (2.2) Release Details:
>
> We have now updated the cover page to include a (placeholder) link to the repository. In the camera-ready version (where anonymity is no longer an issue), this shall be the actual link to the software repository and license. That way, the official “readme” and “requirements.txt” (containing information about package versions and dependencies) will be one click away. (Since Clairvoyance will remain under active development and community engagement, these details may need to be updated rapidly, so a direct link to the software repository is easier to maintain than a static Appendix entry).

---

> ### Author Response · Authors · 2020-11-19
> **Response to Reviewer #3 [Part 1/4]**
>
> ---
>
> Thank you for your thoughtful comments and suggestions. We give answers to each in turn, as well as pointing out corresponding updates to the revised manuscript (additions are made in *blue*).
>
> ---
>
> **(1) Background and References for Experiments**
>
> (1.1) Describing and Justifying Experiments:
>
> We agree that---in describing the illustrative examples---the paper would benefit from a more thorough presentation of background information and justification.
>
> In the revised manuscript, we have now included a (new) Appendix B: “Background References for Experiments”, which tabulates exemplary references to original papers for the following details on each modeling question (e.g. “clinical deterioration of ward patients”), as well as---where appropriate---pointers to the relevant sections in those papers that we follow.
>
> - original works describing and justifying the modeling task,
>
> - original description/documentation for the dataset used, including
>
> - details on data collection, coverage, ethics, etc.,
>
> - the procedure for initial data cleaning/selection that we followed, as well as
>
> - theoretical background on the Clairvoyance pathway involved in the modeling question.
>
> (1.2) Details on Processing, Evaluation, etc:
>
> Thank you for pointing out the benefit of additional clarity with respect to how our experiments are conducted---even more so since reproducibility and transparency are central to our thesis.
>
> To this end, we now provide *fully worked examples* of Clairvoyance in actual usage, which directly demonstrates how an experiment is exactly run, as well as further highlighting the simplicity and standardization of coding with the pipeline abstraction.
>
> (a) In the (new) Appendix E in the revised manuscript, we now provide a worked example of using the *Clairvoyance pipeline* to train and use a model (for this, we use the predictions pathway). This takes the form of a detailed, step-by-step walk-through of the entire pipeline and its components, with fully-executable code, comments, and accompanying descriptions where appropriate.
>
> (b) In the (new) Appendix F in the revised manuscript, we now provide a worked example of using the *optimization interface* to perform stepwise model selection (for this, we use the treatment effects pathway for variety). This also takes the form of executable, commented code, organized as in a standard experiment---using a “main” function wrapper with top-level arguments.
>
> (c) In the (new) Appendix G in the revised manuscript, we now provide an example of how a generic *wrapper class* should be written for the purpose of integrating an external model/algorithm that is not already implemented in the current version of Clairvoyance. Specifically, we show an example of how a classical time-series prediction model (ARIMA) can be easily integrated.
>
> In the main text of the revised manuscript, pointers to all of these worked examples are now included in a dedicated “focus box” at the end of Section 4 with its own subtitle: “Worked Examples”. Furthermore, this also includes references to the interactive tutorials (using Jupyter notebooks) and top-level API code with examples of pathways and optimizations, that are part of the software repository.
>
> Putting it all together, we believe that the combination of the following elements now adequately demonstrates the motivation, contribution, and ease of adoption of the Clairvoyance pipeline: Section 4 (high-level motivations and medically relevant use cases), Appendix B (background and original references), Appendix D (low-level experiment details), and Appendices E, F, G (fully-worked/executable examples).

---

> ### Author Response · Authors · 2020-11-25
> **Dear Reviewer #3**
>
> ---
>
> We are sincerely grateful for your time and energy in the review process. Please let us know if there are any leftover concerns--- We would be happy to do our utmost to address them.
>
> Thank you!
>
> Paper3220 Authors

---

### Official Review · AnonReviewer2 · 2020-10-31
**Exciting and useful pipeline; concerned that the paper doesn't do justice to the authors' work**

**Rating:** 5
**Confidence:** 2

**Review:**

In this paper, the authors showcase a pipeline intended to standardize and industrialize AI model development and testing for medical time series.

It is very clear that the authors have put in a tremendous amount of work in building their pipeline. As someone who works on ML for healthcare, I appreciate it and look forward to using such pipelines.

As far as the *paper* is concerned, however, I’m not sure if the paper (as it is written) and the venue (ICLR) are a good fit. For papers describing such frameworks, I’d prefer to read about:
1. pipeline design and tradeoffs: why did the team make the decisions they did . For e.g. why did they decide to design the API interface the way they did, why did they decide to offer some ML techniques over others, what are the tradeoffs between standardization and flexibility for using such a pipeline, when would a user use Clairvoyance over starting fresh.
2. benefits of using this pipeline over other pipelines or no pipeline: the authors benchmarked their pipeline’s performance over off-the-shelf ML models for some tasks, which is great! It would also be good to see the benefits of using Clairvoyance in terms of time to setup, time to train, etc.
3. user interviews, feedback, and adoption to demonstrate how quickly new users can learn this framework and the benefits they observe while using it

I look forward to hearing back from the authors and I’m open to changing my score.

---

> ### Author Response · Authors · 2020-11-19
> **Response to Reviewer #2 [Part 5/5]**
>
> ---
>
> **References** (numbering as continued from revised manuscript)
>
> [99] ICLR 2020: Novak et al., "Neural Tangents: Fast and Easy Infinite Neural Networks in Python".
>
> [100] ICLR 2020: Osband et al., "BSuite: Behavior Suite for Reinforcement Learning".
>
> [101] ICLR 2019: Schneider et al., "DeepOBS: A Deep Learning Optimizer Benchmark Suite".
>
> [102] ICML 2020: Goyal et al., "PackIt: A Virtual Environment for Geometric Planning".
>
> [103] ICML 2020: Hu et al., "EXTREME: A Massively Multilingual Multi-task Benchmark for Evaluating Cross-lingual Generalization".
>
> [104] ICML 2019: Bansal et al., "HOList: An Environment for Machine Learning of Higher-Order Theorem Proving".
>
> [105] NeurIPS 2020: Schoenholz et al., "JAX, M.D.: A Framework for Differentiable Physics".
>
> [106] NeurIPS 2019: Tran et al., "Bayesian Layers: A Module for Neural Network Uncertainty".
>
> [107] NeurIPS 2019: Park et al., "Park: An Open Platform for Learning-Augmented Computer Systems".
>
> [108] NeurIPS 2019: Paszke et al., "PyTorch: An Imperative Style, High-Performance Deep Learning Library".
>
> [109] NeurIPS 2017: Paszke et al., "Automatic differentiation in PyTorch".

---

> ### Author Response · Authors · 2020-11-19
> **Response to Reviewer #2 [Part 4/5]**
>
> ---
>
> **(3) User Adoption and Feedback**
>
> Thank you for pointing out the value in measuring engagement and feedback from the community. We completely agree that this an important step in the development cycle of any software framework.
>
> First, however, we would like to point out that the development cycle (as pertains presentation/writing) generally proceeds in two broad stages. (a) In the first instance, the newly developed platform/framework is presented in a “proposal” paper that is part software description and part call-to-arms for the overarching goal. (b) Then, after development matures and community feedback is incorporated over a period of time, a “capstone” paper presents the final design philosophy and description of user engagement/adoption. In this present paper, we are striving to accomplish “Stage (a)” as part of our effort to disseminate and encourage community participation.
>
> For a random selection of similar “Stage (a)” efforts in the machine learning (main conference) community in the past two years, we refer to [99--101] for ICLR, [102--104] for ICML, and [105--107] for NeurIPS. Note that although the specific ML subfields are different from ours, these all similarly propose high-level API software packages and/or standardized environments to enable rapid prototyping and ease of development, as well as to encourage encourage systematic and reproducible validation. (For a classic example of this “proposal + capstone” pattern of development and presentation, see [108--109]; note that the study on user adoption is reported in the capstone paper).
>
> Second, we do have a fully-developed agenda for *community engagement* in execution, which would lead us towards “Stage (b)” in time. While we are necessarily brief on the following details (due to the requirement for anonymity during the review process), this should give an adequate idea of outreach:
>
> - Our lab introduced this framework/initiative in May of this year in an official announcement, accompanying technical description, and publicly available source code.
>
> - In a keynote address at a workshop in a prominent ML conference in July, this framework was introduced, and participants were encouraged to download, try out, and give feedback on our software.
>
> - In August, a piece of long-form content was published by the team, highlighting the framework and calling for readers to visit the software repository.
>
> Moreover, from September to the present (and continuously going forward), we introduced the framework through a series of our lab’s hour-long online “engagement sessions” for machine learning and healthcare researchers and students. Specifically:
>
> - In September, a general introduction to the pipeline abstraction was given, while participants were encouraged to visit our software repository and try the framework in their own projects.
>
> - In October, our engagement session featured a 10-minute section specifically dedicated to explaining the mission of the initiative, as well as the goals of the framework.
>
> - Most recently, in November our engagement session focused extensively on the framework--including 16 minutes of tutorials on how to use the software, and how to add new algorithms to the framework, as well as fielding numerous questions on the pipeline during the subsequent Q&A session.
>
> In time, these initiatives will lead us towards more mature community engagement, whereupon we will be equipped to measure and report user interviews, feedback, and adoption patterns (i.e. “Stage (b)”). Finally, we note that this project was precisely born out of collaborations between a team of ML researchers and medical professionals: With university clinicians as developers/authors, our central design goal is always to provide for easy adoption in realistic usage scenarios.
>
> ---
>
> With our clarifications and revisions, we hope that we have addressed your concerns. Thank you for your kind consideration.

---

> ### Author Response · Authors · 2020-11-19
> **Response to Reviewer #2 [Part 3/5]**
>
> ---
>
> **(2) Benefits of Using Clairvoyance**
>
> (2.1) Time to Setup:
>
> Further to the quantitative results in Section 4, we agree that additional indication regarding the *ease-of-use* of Clairvoyance would better clarify its advantage---over starting from scratch.
>
> Now, it is difficult (if not impossible) to accurately quantify any notion of “time savings” involved in adopting Clairvoyance for use during research and development of any medical time-series project. However, we do believe that providing *fully worked examples* of Clairvoyance in actual usage shall more than adequately demonstrate the significant benefits of simplicity and standardization in any actual coding work required.
>
> (a) In the (new) Appendix E in the revised manuscript, we now provide a worked example of using the *Clairvoyance pipeline* to train and use a model (for this, we use the predictions pathway). This takes the form of a detailed, step-by-step walk-through of the entire pipeline and its components, with fully-executable code, comments, and accompanying descriptions where appropriate.
>
> (b) In the (new) Appendix F in the revised manuscript, we now provide a worked example of using the *optimization interface* to perform stepwise model selection (for this, we use the treatment effects pathway for variety). This also takes the form of executable, commented code, organized as in a standard experiment---using a “main” function wrapper with top-level arguments.
>
> (c) In the (new) Appendix G in the revised manuscript, we now provide an example of how a generic *wrapper class* should be written for the purpose of integrating an external model/algorithm that is not already implemented in the current version of Clairvoyance. Specifically, we show an example of how a classical time-series prediction model (ARIMA) can be easily integrated.
>
> In the main text of the revised manuscript, pointers to all of these worked examples are now included in a dedicated “focus box” at the end of Section 4 with its own subtitle: “Worked Examples”. Furthermore, this also includes references to the interactive tutorials (using Jupyter notebooks) and top-level API code with examples of pathways and optimizations, that are part of the software repository.
>
> From these detailed examples, the benefits with respect to easy adoption, rapid prototyping, and extensibility in incorporating new techniques should become apparent. Note that these worked examples are also relevant in support of points 1.1(a,b,c) above).
>
> As a final clarification, allow us to reiterate that the proposed framework is the first of its kind, and there are no other frameworks that provide an end-to-end, unified structure to the *entire* time-series workflow (see Section 4: “Related Work”, and Table 1: “Clairvoyance and Comparable Software”). While there is existing software providing *implementations* of individual models and optimization algorithms, there is no comparable *framework-level* software that makes sense to “juxtapose” with (especially in the medical setting)---in fact, this is precisely what motivated developing/disseminating Clairvoyance in the first place.
>
> (2.2) Time to Train:
>
> At risk of belaboring the point, Clairvoyance is a pipeline abstraction (i.e. framework), not a specific algorithm (i.e. implementation). Therefore the “time to train” depends entirely on (a) which individual techniques are ultimately selected by the user, and (b) the characteristics of the dataset being used. This is also true of the optimization interface; i.e. the time to train depends on the specific optimizer selected. Moreover, all of this depends on (c) the hardware on which the computation is performed.
>
> Importantly, however, we can certainly provide an upper bound on the “typical” time to train as pertains the types of datasets we used (see characteristics and statistics in Table 2) in our experiments, including those performing Bayesian optimization for stepwise model selection on top of the pipeline. Our computations were performed with a single NVIDIA GeForce GTX 1080 Ti GPU, and each experiment took approximately ~24--72 hours. Of course, this duration may be shortened through the use of multiple GPUs in parallel.
>
> In the revised manuscript, this brief note on “Time to Train” is now included at the end of the (new) Appendix C.

---

> ### Author Response · Authors · 2020-11-19
> **Response to Reviewer #2 [Part 2/5]**
>
> ---
>
> (1.2) Tradeoff between Flexibility and Standardization:
>
> Thank you for pointing out the importance of considering this tradeoff. We agree that contextualizing our mission along this tradeoff would better clarify our motivation and contribution.
>
> First, we note that the modularity of the pipeline no doubt enforces a specific form of “standardization” of workflows. However, there are two possible senses of “flexibility”: (a) in the freedom to customize pipelines within the proposed abstraction, and (b) in the generality of the pipeline abstraction itself. Briefly, with respect to (a), it is easy to see that our design emphasis on encapsulation and extensibility means that pipelines are flexibly customized as required---both by mixing and matching component models as needed, as well as by easily incorporating novel methods through minimal wrapper classes. In this sense, then, Clairvoyance seeks to maximize standardization and flexibility *in parallel*: Unless there is a compelling reason for entangling two or more steps in the pipeline, there is little disadvantage to adopting the Clairvoyance workflow instead of starting fresh.
>
> As for (b), first allow us to clarify that our central thesis revolves around bettering the *research-and-development* workflow for medical time-series. At slight risk of generalization, this---importantly---contrasts with the downstream goal of *productionizing* mature architectures for deployment (which is firmly beyond the scope of our mission). In the former, our focus on rapid prototyping and reproducible experimentation means that “standardization” is paramount, and the proposed pipeline abstraction is firmly positioned as such. In the latter, of course, this second notion of “flexibility” becomes much more important to balance against: Various application-/preference-specific considerations arise from the practical needs of deployment and integration, such as handling data streams, patient privacy, algorithmic fairness, or compliance with government regulations. In this sense, a development team looking to address such concerns would need to more carefully weigh the *tradeoff* between flexibility and standardization.
>
> In the revised manuscript, this explanation of the tradeoff is now included in a dedicated “focus box” under Section 2 with its own subtitle: “Flexibility vs. Standardization”.
>
> (1.3) Choice of Built-in Techniques:
>
> We agree a discussion of our choice of built-in techniques would better position the focus of our work.
>
> By way of preface, allow us to reiterate our “Pipeline First” focus (see point 1.1(a) above), especially in the context of medical settings: Rather than (re-)implementing every time-series model in existence, our primary contribution is in unifying all three key pathways in a patient’s healthcare lifecycle (i.e. predictions, treatments, and monitoring tasks; see Section 2: “The Patient Journey”) through a single end-to-end pipeline abstraction---for which Clairvoyance is the first (see Table 1: “Clairvoyance and Comparable Software”).
>
> For the predictions pathway, while there is a virtually infinite variety of time-series models in the wild, we choose to include standard and popular classes of *deep learning* models, given their ability to handle large amounts and dimensions of data, as well as the explosion of their usage in medical time-series studies (see e.g. virtually any of the paper references in Section 1). For both the treatment effects and active sensing pathways, there is much less existing work available; for these, we provide state-of-the-art models (e.g. CRN, R-MSN, ASAC, DeepSensing) implemented exactly as given in their original research papers.
>
> With that said, as noted throughout, recall that all component modules (including the various other pipeline components) are easily *extensible*: For instance, if more traditional time-series baselines from classical literature were desired for comparison purposes, existing algorithms from [73--79] can be integrated into Clairvoyance by using simple wrapper classes, with little hassle (for an explicit demonstration of this, see also our response to point (2.1) below for reference to the newly included example in Appendix G).
>
> In the revised manuscript, this discussion on built-in implementations is now included in a (newly added) Appendix C. In addition, in the main text, pointers to Appendix C and Appendix G are now included in a small box (due to space constraints) at the end of Section 2 with subtitle: “Implementations and Extensions”.

---

> ### Author Response · Authors · 2020-11-19
> **Response to Reviewer #2 [Part 1/5]**
>
> ---
>
> Thank you for your thoughtful comments and suggestions. We give answers to each in turn, as well as pointing out corresponding updates to the revised manuscript (additions are made in *blue*).
>
> ---
>
> **(1) Pipeline Design and Tradeoffs**
>
> (1.1) Considerations in Design of API:
>
> We agree that---in presenting a software framework---the paper would benefit from an explicit description of the considerations driving the API’s design. Our design philosophy is based on the authors’ experience in prototyping and developing real-world collaborative research projects in clinical time-series settings:
>
> (a) Pipeline First, Models Second: Our first emphasis is on *reproducibility*: The process of engineering and evaluating complete medical time-series workflows needs to be clear and transparent. Concretely, this manifests in the strict “separation of concerns” enforced by the high-level API of each component module along the pipeline (see e.g. Figure 3 for illustration). With the ProblemMaker and PipelineComposer as first-class objects, the central abstraction here is the *pipeline* itself; the intricacies and configurations of individual models (e.g. a specific deep learning temporal imputation method) are limited to within each component module.
>
> (b) Be Minimal and Unintrusive: Our second emphasis is on *standardization*: While workflow development needs to be unified and systematic, learning to use the framework should be intuitive. Concretely, this manifests in the API’s adherence to the existing and popular “fit-transform-predict” paradigm (see e.g. sklearn) in all component modules---both *along* the pipeline steps, as well as *across* the pathways that define the patient’s healthcare lifecycle. This enables easy adoption and rapid prototyping---qualities that are paramount given the degree of collaborative research and cross-disciplinary code-sharing required in healthcare-related research.
>
> (c) Encourage Extension: Our third emphasis is on *extensibility*: Given that novel methods are proposed in the ML community every day, the pipeline components should be easily extensible to incorporate new algorithms. Concretely, this manifests in the *encapsulated* design for models within each component module: Specifically, in order to integrate a new component method (e.g. from another researcher’s code, or from an external package) into the framework, all that is required is a simple wrapper class that implements the “fit”, “predict”, and “get_hyperparameter_space” methods; likewise, for an optimization agent, all that is required is to expose an “optimize” method.
>
> In the revised manuscript, these points pertaining to design philosophy are now explained in a dedicated “focus box” under Section 2 with its own subtitle: “Key Design Principles”.

---

> ### Author Response · Authors · 2020-11-25
> **Dear Reviewer #2**
>
> ---
>
> We are sincerely grateful for your time and energy in the review process.
>
> In light of our responses (Nov 19) and revisions (Nov 19), we would appreciate if the reviewer kindly let us know of any leftover concerns in the very limited time remaining. We would be happy to do our utmost to address them.
>
> Thank you!
>
> Paper3220 Authors

---

### Author Response · Authors · 2020-11-19
**Revised Paper**

---

We thank the reviewers again for their thoughtful comments.

We have updated the paper to more clearly and accurately reflect our specific positioning and contributions. Broadly, this includes (a.) fully-worked examples with code, (b.) additional background on experiments, (c.) specifics regarding software design principles, and (d.) miscellaneous clarifications. We are grateful for all questions and suggestions, which improve the clarity and thoroughness of our positioning and presentation. All additions to the manuscript are made in *blue*.

---

**(a.) Fully-worked Examples with Code**

[Reviewers 1, 2, 3, and 4]:

- **New Appendix E** (“Worked Example: Using the Full Pipeline”): We now provide a worked example of using the *Clairvoyance pipeline* to train and use a model (for this, we use the predictions pathway). This takes the form of a detailed, step-by-step walk-through of the entire pipeline and its components, with fully-executable code, comments, and accompanying descriptions where appropriate.

[Reviewers 1, 2, and 3]:

- **New Appendix F** (“Worked Example: Using the AutoML Interface”): We now provide a worked example of using the *optimization interface* to perform stepwise model selection (for this, we use the treatment effects pathway for variety). This also takes the form of executable, commented code, organized as in a standard experiment---using a “main” function wrapper with top-level arguments.

[Reviewers 1, 2, and 3]:

- **New Appendix G** (“Extensibility: Example Wrapper Class”): We now provide an example of how a generic *wrapper class* should be written for the purpose of integrating an external model/algorithm that is not already implemented in the current version of Clairvoyance. Specifically, we show an example of how a classical time-series prediction model (ARIMA) can be easily integrated.

[Reviewers 1, 2, and 3]:

- **New Focus Box** (“Worked Examples”): In Section 4 of the revised manuscript, pointers to all of these worked examples are now included in a dedicated “focus box”. Furthermore, this also includes references to the *interactive tutorials* (using Jupyter notebooks) and *top-level API code* with examples of pathways and optimizations, that are part of the software repository.

---

**(b.) Additional Background on Experiments**

[Reviewers 1 and 3]:

- **New Appendix B** (“Background References for Experiments”). We now provide a complete tabulation of exemplary references to original papers for the following details on each modeling question examined in the Section 4 experiments (e.g. “clinical deterioration of ward patients”), as well as---where appropriate---pointers to the relevant sections in those papers that we follow. These references include original works describing and justifying the modeling task; original description/documentation for the dataset used, including details on data collection, coverage, ethics, etc.; the procedure for initial data cleaning/selection that we followed; as well as theoretical background on the Clairvoyance pathway involved in the modeling question. Note that this tabulation of details is *in addition* to the medical motivation provided in Section 4, the low-level experiment details in Appendix D, as well as the worked-out code examples now provided in Appendices E, F, and G.

---

> ### Author Response · Authors · 2020-11-19
> **Revised Paper (continued)**
>
> ---
>
> **(c.) Specifics Regarding Software Design Principles**
>
> [Reviewer 2]:
>
> - **New Focus Box** (“Key Design Principles”): In Section 2 of the revised manuscript, the framework’s design philosophy is now explained in detail, including: (a) Pipeline First, Models Second (emphasis is on *reproducibility*), and how it concretely manifests in the framework’s design. (b) Be Minimal and Unintrusive (emphasis is on *standardization*), and how it concretely manifests in the framework’s design. (c) Encourage Extension (emphasis is on *extensibility*), and how it concretely manifests in the framework’s design.
>
> [Reviewer 2]:
>
> - **New Focus Box** (“Flexibility vs. Standardization”): In Section 2 of the revised manuscript, the intended usage of Clairvoyance is clarified in a discussion about flexibility and standardization. In particular, we reiterate our focus on the *research-and-development* workflow for medical time-series (in which flexibility and standardization go hand-in-hand), which contrasts with the downstream goal of *productionizing* mature architectures for deployment (in which a tradeoff exists, but which is firmly beyond the scope of our mission).
>
> [Reviewers 1, 2, and 3]:
>
> - **New Focus Box** (“Implementations and Extensions”): In Section 2 of the revised manuscript, a pointer is now included to a (new) Appendix C (“Choice of Built-in Techniques”), where we provide a discussion and justification of the initial choice of algorithms included in the current version of the framework. In addition, a pointer is now included to the (new) Appendix G (“Extensibility: Example Wrapper Class”, mentioned above), where we provide an example of how an external model can be easily integrated into the pipeline.
>
> ---
>
> **(d.) Miscellaneous Clarifications**
>
> [Reviewer 1]:
>
> - **New Appendix I** (“Glossary of Acronyms”). We now provide a list of all potentially lesser-known acronyms. Where appropriate, we also provide the original citation to the paper in which the acronym is first defined.
>
> [Reviewers 1, 3]:
>
> - **Package Details**: We have now updated the cover page to include “Python” in front of the (placeholder) link to the repository. In the camera-ready version (where anonymity is no longer an issue), this shall be the actual link to the software repository and license. That way, the official “readme” and “requirements.txt” (containing information about package versions and dependencies) will be one click away. (Since Clairvoyance will remain under active development and community engagement, these details may need to be updated rapidly, so a direct link to the software repository is easier to maintain than a static Appendix entry).
>
> [Reviewer 4]:
>
> - **Grammar**: We have now revised the problematic sentence to resolve the grammatical issue.
>
> ---
>
> With our clarifications and revisions, we hope that we have addressed the reviewers' concerns. Thank you for your kind consideration.

---

### Comment · ~Jason_X_Dou1 · 2022-01-08
**amazing tool and initiative**

I find this tool/paper online and love it. I would say it's among the rare papers in the "top ML conferences" that are really "practical and useful". I would like to see more papers like this to be published instead of paper addressing "fancy problems" with tons of "seemingly fancy" math/proofs.

---

### Decision · Program_Chairs · 2021-01-07
**Final Decision**

**Decision:**

Accept (Poster)

**Comment:**

The paper addresses a pressing problem for applications involving clinical time series and introduce a pipeline that handle many of the issues pertaining to data preprocessing.

An important contribution is the software that makes the processing more seamless, which will, without a doubt, be useful to the community given the need for reproducibility.

The authors have responded suitably to reviewer comments with the main 'leftover criticism' being that such a paper may not be the best fit for ICLR. This isn't a typical paper. However, something that introduces this level of automation and flexibility in handling time series has not been presented at this conference (or other ML conferences) to the best of my knowledge. It seems it could work in conjunction (as opposed to competing) with any new time series models/techniques that may be introduced.